# CURVALID: A GEOMETRICALLY-GUIDED ADVERSARIAL PROMPT DETECTION

## ABSTRACT

Adversarial prompts that can jailbreak large language models (LLMs) and lead to undesirable behaviours pose a significant challenge to the safe deployment of LLMs. Existing defenses, such as input perturbation and adversarial training, depend on activating LLMs' defense mechanisms or fine-tuning LLMs individually, resulting in inconsistent performance across different prompts and LLMs. To address this, we propose CurvaLID, an algorithm that classifies benign and adversarial prompts by leveraging two complementary geometric measures: Local Intrinsic Dimensionality (LID) and curvature. LID provides an analysis of geometric differences at the prompt level, while curvature captures the degree of curvature in the manifolds and the semantic shifts at the word level. Together, these tools capture both prompt-level and word-level geometric properties, enhancing adversarial prompt detection. Unlike traditional approaches that rely on token-level LID, we introduce PromptLID, which calculates LID at the prompt level to better capture the geometric properties of text prompts within the adversarial subspace. Additionally, we propose TextCurv to further analyze the local geometric structure of prompt manifolds by calculating the curvature in text prompts. CurvaLID achieves over 0.99 detection accuracy, effectively reducing the attack success rate of advanced adversarial prompts to zero or nearly zero. Importantly, CurvaLID provides a unified detection framework across different adversarial prompts and LLMs, as it achieves consistent performance regardless of the specific LLM targeted.

## 1 INTRODUCTION

The safety of Large Language Models (LLMs) has gained significant attention due to their deployment in many real-world applications. Research into adversarial prompts that threaten LLM safety has grown, along with the development of defenses. Adversarial training and prompt engineering have been adapted to defend LLMs. Input perturbation techniques, such as IntentionAnalysis (Zhang et al., 2024a) and SmoothLLM (Robey et al., 2023), depend on how the prompt appears after perturbation and whether the modified prompt can successfully trigger the self-defense mechanisms of the targeted LLM. The effectiveness of these techniques varies depending on the extent of the perturbation and the LLM being targeted. Adversarial training methods require fine-tuning of each model. Moreover, they often face limitations when scaling to larger models. For example, Latent Adversarial Training (LAT) is limited to LLMs with fewer than 10 billion parameters (Sheshadri et al., 2024). Both adversarial training and prompt engineering defenses are dependent on the targeted LLMs. As a result, they fail to guarantee robust protection across different adversarial prompts and models.

An alternative defense method is adversarial prompt detection, which is both efficient and practical. In this approach, practitioners can reject queries that contain adversarial prompts. Adversarial input detection has been extensively studied in the context of image classification tasks (Dong et al., 2020; Machado et al., 2021; Bajaj & Vishwakarma, 2024; Liu et al., 2024). For instance, Ma et al. (2018) demonstrated that Local Intrinsic Dimensionality (LID) can effectively characterize an adversarial subspace and enable efficient adversarial input detection. They leveraged image-level representation vectors to estimate LID and distinguish adversarial examples from clean inputs. However, a fundamental difference in LLMs lies in the fact that inputs are represented as sequences of words. To the best of our knowledge, no prior work has investigated LID in the context of adversarial prompt detection for LLMs. Yin et al. (2024) explored using LID to assess the truthfulness of LLM responses

by estimating LID with token-level representations. We demonstrate that token-level LID is ineffective for adversarial prompt detection because LID values for benign and adversarial prompts often overlap significantly, primarily due to shared nearest neighbors being dominated by stop words and punctuation. Even after removing stop words and punctuation, which reduces the standard deviations, the average LID values for benign and adversarial prompts remain similar, making token-level LID still ineffective for distinguishing between them.

To address these limitations, we introduce CurvaLID, an algorithm designed to classify benign and adversarial prompts by leveraging two complementary geometric measures: LID and curvature. Instead of using token-level representations, CurvaLID utilizes PromptLID, a prompt-level LID analysis that quantifies and characterizes the distinct local subspace where adversarial prompts reside. We train a simple convolutional neural network (CNN) on clean prompts to extract these prompt-level representations, enabling the distinct LID of adversarial prompts to be identified during testing. Unlike previous work (Ma et al., 2018), which relies on the specific victim model for detection, CurvaLID detects adversarial prompts across different LLMs using a separate CNN for prompt representation extraction. This independence from specific LLMs ensures broader applicability and allows CurvaLID to provide consistent and universal defense across different LLMs. Furthermore, we introduce TextCurv, a novel mathematical framework that captures the geometric properties of text prompts through curvature analysis, focusing on the degree of curvature in the manifolds and the semantic shifts at the word level. Together, PromptLID and TextCurv provide complementary information—PromptLID captures prompt-level geometric differences, while TextCurv reveals local structural differences at the word level. While PromptLID achieves great accuracy, when combined with TextCurv, CurvaLID surpasses 0.99 classification accuracy, significantly outperforming state-of-the-art (SOTA) defenses on various adversarial prompts, including PAIR (Chao et al., 2023), RandomSearch (Andriushchenko et al., 2024), and AmpleGCG (Liao & Sun, 2024). CurvaLID consistently performs across all LLMs by detecting adversarial prompts independently, without depending on specific model architectures as many SOTA defenses do. CurvaLID is highly efficient, requiring only 15 minutes of training on a single Nvidia H100 GPU, compared to SOTA adversarial training methods that can take up to 16 hours on an 8-node A100 setup (Mazeika et al., 2024) or nearly 12 hours on a single A100 or H100 GPU (Sheshadri et al., 2024).

Our main contributions can be summarized as follows:

- We devise PromptLID, a prompt-level LID estimation method that enhances adversarial prompt detection, and effectively addresses the limitations of conventional token-level LID estimation. By leveraging prompt-level representations learned by a simple CNN, we can effectively capture the distinct characteristics of clean and adversarial prompts in their local intrinsic dimension.

- We propose TextCurv, a novel method for calculating curvature in text prompts. TextCurv captures local geometric properties by focusing on the semantic shift between consecutive words and the magnitude of this change. It effectively distinguishes between harmless and adversarial prompts, providing a robust classification capability.

- We introduce CurvaLID that utilizes both PromptLID and TextCurv to distinguish benign and adversarial prompts. Its performance is independent of the underlying LLM and offers consistent and efficient adversarial detection.

## 2 RELATED WORK

This section reviews prior research on adversarial attacks and defense mechanisms for LLMs, highlighting their objectives, underlying logic, and key characteristics.

**Adversarial attacks on LLM.** Adversarial attacks on LLMs involve crafted inputs designed to manipulate models into generating harmful content, like offensive language or dangerous instructions (Zou et al., 2023). These attacks range from a single input (zero-shot) to more complex, continuous dialogue scenarios (multi-shot) (Shen et al., 2023; Dong et al., 2023; Wang et al., 2023). This research focuses on zero-shot text prompt attacks, including techniques like text perturbation that adds gibberish or subtly alters input wording and social-engineered prompts that trick LLMs into harmful behavior (Zou et al., 2023; Schwinn et al., 2023; Chu et al., 2024).

**Adversarial defenses for LLM.** There are three primary defenses against adversarial attacks on LLMs: input preprocessing, prompt engineering, and adversarial training. Input preprocessing perturbs inputs to disrupt adversarial prompts but may also affect benign ones, with its effectiveness varying depending on the extent of the perturbation and the targeted LLM(Cao et al., 2023; Robey et al., 2023; Yung et al., 2024). Prompt engineering enhances the LLM's self-defensive capabilities by adding prompts that help expose harmful intent, though its effectiveness varies with different models (Zhang et al., 2024a; Phute et al., 2023; Zhang et al., 2024b). Finally, adversarial training strengthens the LLM's ability to reject harmful prompts by exposing it to adversarial scenarios during training (Xu et al., 2024b; Jain et al., 2023). However, this approach requires fine-tuning the LLM, making its success dependent on the specific model being protected, with most adversarial training methods confined to white-box models. Moreover, these methods generally exhibit varying effectiveness across different adversarial prompts and LLMs (Sheshadri et al., 2024; Ziegler et al., 2022; Ganguli et al., 2022). Note that our paper belongs to the field of adversarial prompt detection. In contrast to our approach, existing methods, such as perplexity filtering, rely on LLMs for next-token probability predictions (Hu et al., 2023).

## 3 BACKGROUND AND TERMINOLOGY

This section provides a brief overview of the mathematical definitions of LID and curvature, as well as how they are estimated and utilized in practice.

### 3.1 LOCAL INTRINSIC DIMENSION

Local Intrinsic Dimensionality (LID) measures the intrinsic dimensionality of the local neighborhood around a reference sample (Houle, 2017). Compared to Global Intrinsic Dimension (GID), which measures the degree of the $d$-dimension of the global manifold of a data subset, LID focuses on the local neighborhood of given points. Thus, LID is particularly useful in analyzing high-dimensional data with varying dimensionality across the dataset.

**Definition 3.1** (Intrinsic Dimension). (Houle, 2017) Let $F$ be a real-valued function that is non-zero over some open interval containing $r \in \mathbb{R}$, $r \neq 0$.

The intrinsic dimensionality of $F$ at $r$ is defined as follows, whenever the limit exists:

$$\text{IntrDim}_F(r) \quad \triangleq \quad \lim_{\epsilon \to 0} \frac{\ln\left(F((1+\epsilon)r)/F(r)\right)}{\ln((1+\epsilon)r/r)}.$$

**Definition 3.2** (Local Intrinsic Dimension (LID)). (Houle, 2017) LID is mathematically defined as:

$$\text{LID}_F(r) = \frac{r \cdot F'(r)}{F(r)} = \text{IntrDim}_F(r),$$

where the last equality follows by applying L'Hôpital's rule to the limits. We are interested in a function $F$ that satisfies the conditions of a cumulative distribution function (CDF) and is continuously differentiable at $r$. The local intrinsic dimension at $x$ is in turn defined as the limit, when the radius $r$ tends to zero:

$$\text{LID}_F^* \triangleq \lim_{r \to 0^+} \text{LID}_F(r).$$

We refer to the LID of a function $F$, or of a point $\mathbf{x}$, whose induced distance distribution has $F$ as its CDF. For simplicity, we use the term 'LID' to refer to the quantity $\text{LID}_F^*$.

$\text{LID}_F^*$ is the theoretical definition, and in practice, this quantity needs to be estimated (Levina & Bickel, 2004; Tempczyk et al., 2022). In this work, we use the Method of Moments (MoM) (Amsaleg et al., 2015) due to its simplicity. LID has been applied in many machine learning applications (Gong et al., 2019; Ansuini et al., 2019; Pope et al., 2021; Huang et al., 2024; Zhou et al., 2024a). It is important to note that applying LID estimation in machine learning requires both a distance measure and a set of reference points to select nearest neighbors (NN). In this work, we follow existing works that use the Euclidian distance. The representation of a data point, along with the chosen reference points, significantly influences how LID is interpreted. In adversarial prompt detection, the representation and neighborhood definition directly affect the ability to distinguish between clean and adversarial prompts.

## 3.2 CURVATURE

The intuition of curvature is how quickly a curve changes direction. In geometry, we can visualize curvature through an osculating circle. Curvature can be measured at a given point by fitting a circle to the curve on which the point resides (Kline, 1998). The formal definition is as follows:

**Definition 3.3.** [Curvature measured by osculating circle](Kline, 1998)

The osculating circle for some point $P$ on a curve is defined as the circle tangent to $P$ and passing each other point on the curve. We denote the radius of the osculating circle as $R$. Thus, the curvature $\kappa$ is defined as:

$$\kappa = \frac{1}{R}.$$

For an arbitrary curve, one can extend the concept of curvature to the rate of tangential angular change with respect to arc length, which is known as the Whewell equation (Whewell, 1849). The tangential angular change refers to the change of angle of inclination of tangent at the given point.

**Definition 3.4.** [Curvature by Whewell equation](Whewell, 1849)

Let $s$ be the arc length and tangential angle $\phi$ be the angle between the tangent to point $P$ and the x-axis, for a given point $P$ on a curve. The curvature $\kappa$ is defined as:

$$\kappa = \frac{d\phi}{ds}.$$

Furthermore, in differential geometry, the curvature can be defined as the change of the unit tangent vector with respect to arc length (Shifrin, 2015; O'neill, 2006).

**Definition 3.5.** [Curvature in differential geometry](Shifrin, 2015) Suppose the curve $\alpha$ is parametrized by arc length $s$ and $\mathbf{T}(s)$ is the unit tangent vector to the curve. We define curvature as

$$\kappa(s) = \|\mathbf{T}'(s)\| = \left\|\frac{d\mathbf{T}}{ds}\right\|.$$

Curvature is also defined and utilized in physics. The Frenet-Serret formulas relate curvature to torsion, tangent, normal, and binormal unit vectors (Frenet, 1852). In the Frenet-Serret formulas, the curvature describes the rotational speed along a curve, which is relevant in kinetics and trajectory applications (Huang et al., 2023). Its application extends to autonomous driving, robotics, and quantum computing (Hallgarten et al., 2024; Alsing & Cafaro, 2023; Shabana, 2023).

# 4 GEOMETRIC ANALYSIS AND CURVALID FRAMEWORK

In this work, we aim to develop geometric measures that can effectively characterize benign and adversarial prompts at both the prompt and word levels, and utilize these measurements for adversarial prompt detection. We specifically define our problem statement as follows:

Let $\mathcal{D} = \{(x^i, y^i)\}_{i=1}^n$ be a labeled dataset comprising $n$ i.i.d. samples $x^{(i)}$, where each sample is associated with a label $y^i$. In this context, each input $x^i$ represents a text prompt, with the corresponding label $y^i \in \{0, 1\}$ indicating whether the prompt is benign ($y^i = 0$), or adversarial ($y^i = 1$). Let $\mathcal{M}(x)$ denote the geometric measure applied to a prompt $x$, where $\mathcal{M}(x)$ is composed of two complementary components: the prompt-level measure PromptLID($x$) and the word-level measure TextCurv($x$). These measures are then utilized within an adversarial prompt detection algorithm, formalized as a classification problem where the objective is to minimize the empirical error between the ground-truth labels and the predictions:

$$\arg \min_\theta \mathbb{E}_{(x,y)\in\mathcal{D}}[\ell(h(\mathcal{M}(x)), y)],$$

where $\ell(\cdot)$ denotes the cross-entropy loss function, and $h(\mathcal{M}(x))$ is the classifier applied to the geometric measures $\mathcal{M}(x) = (\text{PromptLID}(x), \text{TextCurv}(x))$.

Next, we formally define PromptLID and TextCurv, detailing how they explore the geometric properties of prompts at the prompt and word levels, respectively. Finally, we provide an overview of CurvaLID, our adversarial prompt detection model.

## 4.1 PROMPTLID: LID ESTIMATION AT THE PROMPT-LEVEL

To characterize the prompt-level geometric properties of benign and adversarial prompts, we propose PromptLID, an LID estimation based on prompt representations obtained from a trained CNN. We first train a model $g$ (CNN) to perform prompt classification. Classification involves learning a function $g : \mathcal{B} \rightarrow \mathcal{Q}$ that maps the input space to the label space $\mathcal{Q}$. In our setup, $\mathcal{Q}$ consists of four distinct labels $\{q_1, q_2, q_3, q_4\}$, each corresponding to one of the four benign datasets we use, namely Orca, MMLU, AlpacaEval, and TQA. For a benign prompt dataset $\mathcal{B} = \{(\boldsymbol{b}, q)^i\}_{i=1}^{n}$, where $\boldsymbol{b}$ is the benign prompt and $q$ is the corresponding dataset label, the model aims to identify which benign dataset each prompt belongs to. The objective function used is categorical cross-entropy, as the task involves multi-class classification. The representation $\boldsymbol{z}_1$, derived from the penultimate dense layer, encodes the prompt as a single vector, which is then used to calculate the PromptLID. The PromptLID is calculated based on the MoM estimation (Amsaleg et al., 2015) of LID on the prompts' representation in $\boldsymbol{z}_1$.

**Definition 4.1** (PromptLID). The PromptLID of a given prompt $\boldsymbol{x}$ is defined as:

$$PromptLID = -k \cdot \frac{\mu_k}{\mu_k - w^k},$$

where $k$ is a locality hyperparameter representing the number of nearest neighbors, $\mu_k$ is the averaged distance from the prompt representation $\boldsymbol{z}_1$ of $\boldsymbol{x}$ to its $k$-nearest neighbors, $w^k$ is the distance to the $k$-th nearest neighbor.

Additionally, we extract the outputs of the convolutional layers, $\boldsymbol{z}_2$ and $\boldsymbol{z}_3$, to calculate TextCurv, a method introduced in the next section, which captures the word-level geometric properties of prompts.

## 4.2 TEXTCURV: CURVATURE AT THE WORD-LEVEL

To characterize the word-level geometric properties of benign and adversarial prompts, we analyze the curvature of word connections. The aim is to have curvature complement PromptLID by analyzing the word-level geometric properties of prompts, effectively identifying nearly all adversarial prompts. Specifically, curvature captures the relationships between words, revealing subtle semantic shifts based on word order, and uncovering local geometric differences between benign and adversarial prompts.

Previous research indicates that CNN activation creates a curved manifold, evidenced by the significantly higher intrinsic dimension estimated by Principal Component Analysis (PCA) compared to the GID estimated by TwoNN on activation data (Abdi & Williams, 2010; Facco et al., 2017; Ansuini et al., 2019). However, there is a lack of findings on the curvature of benign versus adversarial prompts. Therefore, we investigate the curvature differences between these prompts in CNN convolution layers as potential classification features.

Our goal is to establish a definition of text curvature based on existing mathematical definitions, with the curvature capturing semantic shifts according to word sequence and the strength of these shifts. Word sequence plays a crucial role in semantic analysis, helping to accurately capture the local geometric properties of prompts. We focus on the representations of prompts in the convolutional layers of the model $g$ as mentioned in Section 4.1, where the prompt data remains unflattened and in stacked lists of vectors at this stage, which can be viewed as word-level representation. Specifically, we extract the representations $\boldsymbol{z}_2$ and $\boldsymbol{z}_3$ from these convolutional layers for further analysis. This stage is critical, as it is where feature spaces are curved, according to prior research (Ansuini et al., 2019).

To capture semantic shifts between consecutive words in a prompt, we draw on Whewell's equation, where the rate of directional change of a curve is represented by the tangential angular change. In NLP, this angular change is connected to the dot-product formula and cosine similarity, which indicate the semantic similarity or difference between two words (Mikolov, 2013; Levy et al., 2015). We assume that this theory also applies to modern word embeddings like GPT-2 and RoBERTa, and therefore, we define the rate of angular change in text curvature accordingly:

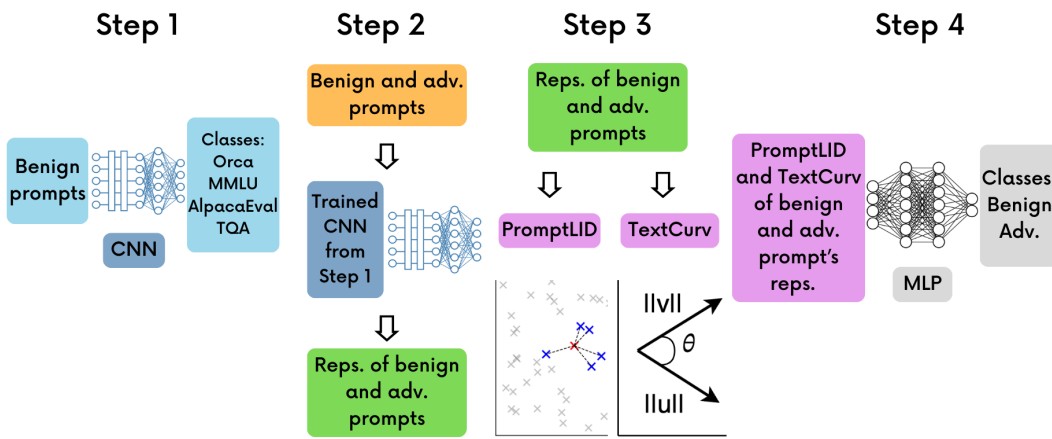

Figure 1: Illustrative diagram for CurvaLID, which classifies benign and adversarial prompts using PromptLID and TextCurv.

**Definition 4.2.** [Text Curvature: Rate of angular change]

For any two consecutive word embeddings, denoted by $\vec{u}$ and $\vec{v}$, the rate of angular change, $d\theta$, is defined as:

$$d\theta = \arccos\left(\frac{\vec{u} \cdot \vec{v}}{\|\vec{u}\|\|\vec{v}\|}\right).$$

However, the rate of angular change alone does not fully capture the semantic shift between words, as it overlooks the magnitude of the shift. In differential geometry, curvature is defined by the rate of change in the tangent vector's direction relative to the change in arc length. When two curves exhibit the same directional change, the curve achieving this change over a shorter arc length has a higher curvature. Similarly, in text curvature, given the same semantic shift as measured by our rate of angular change, the curvature should increase when the semantic change is more substantial.

We focus on word vector magnitudes to capture the degree of semantic shift. Previous research suggests that magnitude reflects the semantic weight carried by each word and the tokenizers' understanding of that word within its context (Schakel & Wilson, 2015; Reif et al., 2019). For example, common words tend to have smaller magnitudes due to their frequent use and limited semantic significance (Schakel & Wilson, 2015). Instead of summing vector norms to measure distance changes in curvature—which may seem intuitive and consistent with geometric principles—we sum the inverses of the vector norms. This approach is driven by the hypothesis that larger vector norms signify greater semantic importance, meaning that curvature should be inversely proportional to vector norms, capturing the larger semantic shifts between words.

**Definition 4.3.** [Text Curvature] For any two consecutive word embeddings, denoted by $\vec{u}$ and $\vec{v}$, the text curvature, denote by TextCurv is defined as:

$$TextCurv = \frac{d\theta}{\frac{1}{\|\vec{u}\|} + \frac{1}{\|\vec{v}\|}}.$$

The supplementary mathematical proof can be found in A.1.2.

### 4.3 CURVALID: ADVERSARIAL PROMPT CLASSIFICATION BY CURVATURE AND LID

CurvaLID classifies and filters adversarial prompts before they reach LLMs, thus ensuring their security. Since CurvaLID operates independently of LLMs, it provides a unified defensive performance across all LLMs. This differentiates it from existing SOTA defenses like input perturbation, prompt engineering, and adversarial training, which show varying performance across different adversarial prompts and LLMs. Moreover, CurvaLID's evaluation is straightforward and standardized, avoiding the need for subjective human assessments or reliance on LLM judgments, which can raise robustness concerns (Chen et al., 2024; Raina et al., 2024).

CurvaLID involves four steps (see Figure 1, pseudo code in Appendix A.1.1). In the first step, we use our trained model $g$ (defined in Section 4.1) to classify different types of benign prompts, deriving the normal feature manifold. This is essential for amplifying the geometric and dimensional distinctions between benign and adversarial prompts in subsequent steps. In the second step, we extract the representations of both benign and adversarial prompts from $z_1$, $z_2$, and $z_3$. In the third step, we compute PromptLID and TextCurv. Then in the final step, we train a Multilayer Perceptron (MLP) to classify benign and adversarial prompts using the two mean TextCurv values, and the PromptLID as inputs. The MLP performs binary classification and filters out adversarial prompts before they reach to LLMs. Details on the MLP architecture are included in Appendix A.1.6.

## 5 EXPERIMENTS

This section highlights CurvaLID's performance in classifying benign and adversarial prompts. We demonstrate the limitations of word-level LID due to stop words and punctuation. Lastly, we analyze TextCurv, emphasizing the difference between benign and adversarial prompts. Unless explicitly specified, the general experimental setup is as follows: we use a total of 2,519 testing prompts, comprising 1,200 benign prompts and 1,319 adversarial prompts. For the benign prompts, we randomly sampled 300 prompts from each of the Orca (Lian et al., 2023), MMLU (Hendrycks et al., 2020), AlpacaEval (Li et al., 2023b), and TruthfulQA (TQA) (Lin et al., 2021) datasets. Similarly, for the adversarial prompts, approximately 300 were randomly sampled from each of SAP (Deng et al., 2023a), DAN (Shen et al., 2023), MathAttack (Zhou et al., 2024b), and GCG (Zou et al., 2023).

We have included ablation studies and comparisons to other defenses in Appendix A.2 and A.4 respectively. It is important to note that SOTA defenses, such as input perturbation and adversarial training, are highly dependent on the specific adversarial prompts used and the particular LLMs targeted. In contrast, CurvaLID operates independently of the LLMs, providing consistent defensive performance across various models. This fundamental difference makes direct comparisons with SOTA defenses challenging. Nonetheless, we have attempted to summarize the performance of SOTA defenses and provide a comparative analysis with our method in Appendix A.4.

### 5.1 ADVERSARIAL PROMPT CLASSIFICATION BY CURVALID

This section explores CurvaLID's performance in classifying adversarial prompts, assessing its ability to identify mixed benign and adversarial inputs in real-world conditions. We also compare CurvaLID's detection capabilities with existing defenses to demonstrate its effectiveness.

#### 5.1.1 CLASSIFICATION ON ADVERSARIAL PROMPTS

In this subsection, we demonstrate CurvaLID's effectiveness in accurately classifying benign and adversarial prompts, including both English and non-English adversarial prompts. We compared a total of eight datasets for our experiment, consisting of four benign datasets, namely Orca, MMLU, AlpacaEval, and TQA, and four adversarial datasets, namely SAP, DAN, MathAttack, and GCG. These datasets are the popular and SOTA English prompt datasets for testing the robustness of LLMs. From each dataset, we randomly selected at least 300 prompts, resulting in over 2,500 prompts for our classification experiment. We evaluated the ability of CurvaLID to distinguish between benign and adversarial prompts. Further details about the datasets are in Appendix A.1.7.

**CurvaLID on English adversarial prompts.** The experimental results, summarized in Table 1, demonstrate CurvaLID's high performance. The model achieved an overall accuracy of 0.992, with perfect accuracy of 1.00 (i.e., 100%) in identifying adversarial prompts and 0.984 accuracy in identifying benign prompts. This indicates that CurvaLID effectively reduced the attack success rate of adversarial prompts to zero.

In addition to evaluating CurvaLID on the four main adversarial datasets, which contain a substantial number of prompts, we extended our experiments to include other datasets like RandomSearch, AmpleGCG, Persuasive Attack, AutoDAN, and DrAttack (Chao et al., 2023; Andriushchenko et al., 2024; Liao & Sun, 2024; Zeng et al., 2024; Liu et al., 2023; Li et al., 2024). These datasets often contain prompts that are highly similar to each other or are variants of those in the datasets used in our main experiment. Due to computational constraints and the limited scope of their original

Table 1: Performance metrics for CurvaLID on benign and adversarial datasets.

| Data class | Accuracy by dataset | | | | Accuracy by class | Overall accuracy | F1 score |
|---|---|---|---|---|---|---|---|
| **Benign** | **Orca** | **MMLU** | **AlpacaEval** | **TQA** | **0.984** | **0.992** | **0.992** |
| | 0.968 | 1.000 | 0.983 | 0.986 | | | |
| **Adversarial** | **SAP** | **DAN** | **MathAttack** | **GCG** | **1.000** | | |
| | 1.000 | 1.000 | 1.000 | 1.000 | | | |

Table 2: Performance metrics for CurvaLID on non-English adversarial datasets.

| Adv. Dataset | zh | it | vi | ar | ko | th | bn | sw | jv | Avg |
|---|---|---|---|---|---|---|---|---|---|---|
| Benign Acc. | 0.975 | 1 | 1 | 1 | 1 | 1 | 1 | 0.975 | 1.0 | 0.994 |
| Adv. Acc. | 1 | 0.984 | 0.984 | 1 | 1 | 1 | 0.984 | 1 | 0.984 | 0.993 |
| Overall Acc. | 0.988 | 0.994 | 0.994 | 1 | 1 | 1 | 0.994 | 0.988 | 0.994 | 0.994 |
| F1 Score | 0.987 | 0.994 | 0.994 | 1 | 1 | 1 | 0.994 | 0.987 | 0.994 | 0.994 |

research, we used 100-200 prompts from each dataset. Despite the reduced number of prompts, CurvaLID effectively identified all adversarial attacks with around 0.99 accuracy. Detailed results of these experiments are provided in Appendix A.2.1.

**CurvaLID on non-English adversarial prompts.** We analyzed CurvaLID's effectiveness in detecting non-English adversarial prompts by testing it on nine languages from the MultiJail dataset (Deng et al., 2023b). We randomly sampled 300 prompts from each of the 9 languages in the MultiJail dataset and tested them individually against 400 benign prompts, adjusted to avoid class imbalance. We gathered 100 prompts from each of four benign datasets randomly. Notably, retraining with benign prompts in different languages was unnecessary to maintain high accuracy. As shown in Table 2, CurvaLID achieved an overall accuracy and F1 score of 0.994, highlighting its robust ability to detect adversarial prompts across languages.

### 5.1.2 DETAILED ANALYSIS ON CURVALID

We conducted an extensive analysis of CurvaLID's performance, including experiments with reduced training prompt sizes and training without adversarial prompts.

**Impact of number of prompts on CurvaLID.** We studied the impact of reducing the number of prompts on CurvaLID's performance. We decreased the number of prompts in each dataset to 150 (from 300). Despite this reduction, CurvaLID maintained an overall accuracy of 0.988 and an F1 score of 0.988. Further details of the experimental results can be found in Appendix A.2.2.

**Using CurvaLID in one-class classification problems.** We experimented at modifying step 4 of CurvaLID by replacing the supervised MLP with unsupervised outlier detection methods such as the local outlier factor (LOF) or isolation forest, which do not require training on adversarial prompts. This was done to explore the performance of the model in a one-class classification scenario, where only benign prompts are used for training. Despite the absence of adversarial examples during training, the LOF and isolation forest methods achieved a comparable detection accuracy of around 0.9. Therefore, our framework is task-independent and can be applied to various problem settings. Further details are provided in Appendix A.2.3.

### 5.2 ANALYSIS OF LOCAL AND GLOBAL ID ON BENIGN AND ADVERSARIAL PROMPTS

This subsection addresses the limitations associated with using token-level LID, for classifying benign and adversarial prompts. Adversarial prompts manipulate rarely encountered areas of the feature space using complex words and irregular combinations (Wallace et al., 2019; Ilyas et al., 2019; Ren et al., 2019). Existing work has shown that adversarial perturbation leads the representation to a subspace with distinct local dimensional properties (high LID) (Ma et al., 2018). As a result, we hypothesize that each word in adversarial prompts could lead to a representation with high LID. By utilizing token-level representations and aggregating the LIDs for each prompt by taking the average, it is possible to detect adversarial prompts.

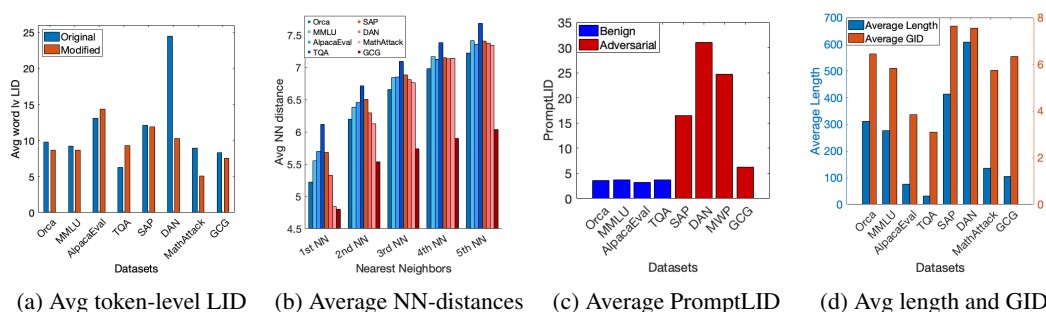

(a) Avg token-level LID     (b) Average NN-distances     (c) Average PromptLID     (d) Avg length and GID

Figure 2: (a) Comparison of average token-level LID for benign and adversarial prompts. The blue bars show the average LID of prompts in their original form, while the red bars show the average token-level LID after removing stopwords and punctuation. (b) Average Nearest Neighbor distances for benign and adversarial datasets. The blue bars represent the average nearest neighbour distances for the benign datasets, while the red bars represent the adversarial dataset. (c) Average PromptLID for benign and adversarial prompts. The benign datasets are in blue while the adversarial datasets are in red. (d) Comparison of average length and average GID across datasets.

Table 3: Top 10 most common nearest neighbors for each dataset. The angle brackets (<>) are used to specify punctuation and newline characters in the tokenizer. The visible space symbol (␣) represents a space preceding a word or punctuation.

| Dataset | Top 10 most common nearest neighbors |
|---|---|
| SAP | ␣to, ␣and, ␣the, <.>, ␣a, ␣of, <,>, ␣<">, ␣that, ␣your |
| DAN | <,>, <.>, <newline>, ␣the, ␣and, ␣to, ␣you, ␣will, ␣not, ␣is |
| MathAttack | ␣the, <,>, ␣of, ␣he, <.>, ␣to, ␣and, ␣she, ␣is, ␣a |
| GCG | ␣text, tto, use, ized, <}>, ␣mar, dt, ␣a, <'>, ␣Guide |
| Orca | ␣the, <,>, <.>, <newline>, ␣and, ␣a, ␣of, ␣to, ␣is, ␣in |
| MMLU | ␣the, <,>, <.>, ␣of, ␣a, ␣to, ␣that, ␣and, ␣in, ␣was |
| AlpacaEval | ␣the, ␣a, ␣to, <,>, ␣of, ␣and, ␣I, <.>, ␣for, <?> |
| TQA | ␣the, <?>, ␣a, ␣you, ␣is, ␣of, ␣that, ␣to, ␣if, ␣in |

**Effectiveness of token-level LID values in separating benign and adversarial prompts.** Our analysis revealed a significant overlap in token-level LID values between benign and adversarial prompts. We examined the average LID at the word level, treating each word representation as a data point with its neighborhood defined by the prompt. This approach mirrors standard natural language processing (NLP) practices, where words are tokenized and converted to embeddings, similar to how pixels are treated as data points in image analysis. Using RoBERTa embeddings, we calculated word-level average LID. As shown in Figure 2a and Appendix A.3.2, the average token-level LID for most datasets are similar and hovers around 10, making it difficult to differentiate between benign and adversarial prompts. The standard deviations of the token-level LID values are also notably high, with AlpacaEval exceeding 100 and DAN surpassing 200. This indicates substantial variability, leading to significant overlap in LID values for benign and adversarial prompts, which diminishes the effectiveness of token-level LID in distinguishing between them.

**Investigating similarity benign and adversarial token-level LIDs.** Next, we investigated the underlying reasons for the similarity in average token-level LID between benign and adversarial prompts. We focused on the average of the first 5 nearest-neighbor distances (NN-distance), which, as shown in Figure 2b, appear similar for both benign and adversarial datasets. We present the top 10 most common nearest-neighbors for different prompts in different datasets in Table 3. It shows that common nearest-neighbors in the representation space were mostly stop words and punctuation. This suggests that word-level LID overlooks the sequential order of text and often relies on conjunctions, articles, and punctuation. These findings suggest that word-level LID is ineffective for detecting adversarial prompts.

**Effectiveness of token-level LID after removing stop words and punctuation.** To explore the significance of stop words and punctuation on the effectiveness of LID, we recalculated word-level

Table 4: Average TextCurv of benign and adversarial datasets across word embedding and CNN layers. Percentages in parentheses represent the increase in TextCurv for adversarial prompts compared to benign prompts.

| Word Embedding | Convolution Layer 1 | | Convolution Layer 2 | |
|---|---|---|---|---|
| | Benign | Adversarial | Benign | Adversarial |
| RoBERTa | 0.626 | 0.881 (+40.7%) | 0.325 | 0.446 (+37.2%) |
| GPT-2 | 0.805 | 1.11 (+37.9%) | 0.389 | 0.546 (+40.4%) |
| BERT | 0.428 | 0.590 (+37.9%) | 0.199 | 0.264 (+32.7%) |
| XLNet | 0.296 | 0.431 (+45.6%) | 0.199 | 0.264 (+32.7%) |
| DistilBERT | 0.386 | 0.557 (+44.3%) | 0.225 | 0.322 (+43.1%) |

LID after excluding these elements. As shown in Figure 2a and Appendix A.3.2, this modification significantly reduces the standard deviation, with a maximum of 2.26, indicating more consistent LID values that are less influenced by stop words and punctuation. However, the difference between the benign and adversarial prompts' LID is not sufficient to discriminate them.

**Analysis on PromptLID.** We analyze PromptLID across benign and adversarial datasets. As shown in Figure 2c, adversarial prompts have a significantly higher average PromptLID (19.472) compared to benign prompts (3.509), highlighting its effectiveness in distinguishing adversarial prompts.

**Effectiveness of global intrinsic dimension in adversarial prompt detection.** Global intrinsic dimension (GID) is another plausible approach for the word-level representation. It can avoid aggregating the LID for each word by assessing the GID for each word within the prompt and output a single value. We use the MLE-based estimate from Tulchinskii et al. (Tulchinskii et al., 2024). However, as shown in Figure 2d, there is no clear difference between benign and adversarial datasets. Instead, GID appears to correlate strongly with prompt length, showing Pearson and Spearman correlation coefficients of 0.92 and 0.98, respectively. We also experimented with the prompt's GID after removing stop words and punctuation, but the results remained similar, with a Pearson correlation coefficient of 0.9, indicating the limitations of GID in effectively detecting adversarial prompts.

### 5.3 CURVATURE ANALYSIS ON BENIGN AND ADVERSARIAL PROMPTS

We analyzed the average TextCurv of adversarial prompts in CurvaLID Step 2. To ensure TextCurv is independent of the embedding model, we conducted curvature analysis using various embeddings, including GPT-2, BERT, XLNet, and DistilBERT (Radford et al., 2019; Devlin, 2018; Yang, 2019; Sanh, 2019). As shown in Table 4, adversarial prompts consistently showed at least 30% higher curvature than benign prompts across all embeddings. This result supports our hypothesis that words in adversarial prompts tend to be more irregular and complex than those in benign prompts.

We also demonstrated that CNN activation significantly amplifies the TextCurv differences between benign and adversarial prompts. Specifically, the mean TextCurv of adversarial prompts is at least 30% higher than that of benign prompts in the representations of both CNN layers. In contrast, when considering only the word embedding, the TextCurv for benign prompts based solely on word embedding is 4.91, compared to 5.42 for adversarial prompts. The difference in TextCurv is much smaller—just 13%, which is less than half of the difference observed in the CNN layers.

## 6 CONCLUSION

In this paper, we introduce CurvaLID, a framework that uses LID and curvature to distinguish benign and adversarial prompts. CurvaLID achieves over 0.99 accuracy and reduces the ASR of the tested adversarial prompts to near zero. Unlike conventional SOTA defenses, such as prompt engineering and input perturbation, CurvaLID operates independently of LLMs, providing consistent performance across all models. We also demonstrate the limitations of word-level LID due to stop words and punctuation. We address this by introducing PromptLID and TextCurv, which respectively explore the geometric properties of prompts at the prompt-level and word-level in text data. Future directions for this work include expanding CurvaLID to spam detection, adding a human feedback system to CurvaLID for optimisation, and testing CuravLID with noisy prompts.

## REPRODUCIBILITY STATEMENT

Details of all hyperparameters and experimental settings are given in Appendix A.1.4, A.1.5, A.1.6, A.1.7, A.2. Pseudo code for LID estimated by Method of Moments can be found in Appendix A.3.1. The experiments were performed on a system equipped with a single Nvidia H100 GPU, 8 CPU cores, and 128 GB of RAM. We provide source code for reproducing the experiments in this paper, which can be accessed here: https://anonymous.4open.science/r/CurvaLID-8DE0/README.md

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

# A APPENDIX

The appendix is organized into four sections. A.1 provides supplementary information about CurvaLID, offering additional details and mathematical proofs. A.2 focuses on ablation studies, presenting experiments to analyze various aspects of CurvaLID's performance. A.3 presents supplementary information on LID analysis. Finally, A.4 evaluates the performance of other SOTA defenses, comparing them to CurvaLID.

## A.1 SUPPLEMENTARY INFORMATION ABOUT CURVALID

This section includes supplementary mathematical proofs for CurvaLID, detailed parameter specifications, and an analysis of its time and space complexity.

### A.1.1 PSEUDOCODE FOR CURVALID

Algorithm A.1 presents the pseudocode for CurvaLID.

---

**Algorithm A.1** CurvaLID

---

**Require:** Datasets $\mathcal{D}_b, \mathcal{D}_a$ (benign and adversarial prompts)
 1: **Step 1: Data Preparation**
 2:     Load datasets $\mathcal{D}_b$ and $\mathcal{D}_a$
 3:     Compute word embeddings $E_b$ and $E_a$ for $\mathcal{D}_b$ and $\mathcal{D}_a$
 4: **Step 2: Preprocessing**
 5:     Pad sequences to uniform length $L_{\max}$
 6:     Standardize embeddings to zero mean and unit variance
 7: **Step 3: Train CNN for Benign Classification**
 8:     Train a CNN $\mathcal{G}$ on $\mathcal{D}_b$ to extract prompt-level representations $\boldsymbol{z}_1$
 9: **Step 4: Compute PromptLID and TextCurv**
10:     Calculate PromptLID on $\boldsymbol{z}_1$
11:     Extract intermediate layer outputs $\boldsymbol{z}_2, \boldsymbol{z}_3$ and calculate TextCurv
12: **Step 5: Train the Detection Model**
13:     Combine PromptLID and TextCurv as features
14:     Train an MLP $\mathcal{H}$ for binary classification of benign vs adversarial prompts

---

### A.1.2 THEORETICAL FOUNDATIONS AND MATHEMATICAL JUSTIFICATION OF TEXTCURV

We begin by addressing the reference of *TextCurv* to Whewell's equation, specifically focusing on how embedding angles relate to tangential angles. Our objective is to demonstrate that the angle between two word embedding vectors corresponds to the difference in their tangential angles (i.e., the numerator in Whewell's equation). To this end, we prove that for two tangent vectors, $\vec{u}$ and $\vec{v}$, in $n$-dimensional Euclidean space, the angle $\theta$ between them is equivalent to the difference in their tangential angles. The following proof establishes this equivalence, showing that the angular difference between two vectors directly corresponds to the difference in their tangential angles.

**Theorem** For two tangent vectors $\vec{u}$ and $\vec{v}$ in $n$-dimensional Euclidean space, the angle $\theta$ between them is equivalent to the difference in their tangential angles.

**Step 1: Angle in $n$-Dimensional Space**

The angle $\theta$ between two vectors $\vec{u}, \vec{v} \in \mathbb{R}^n$ is defined as:

$$\cos \theta = \frac{\vec{u} \cdot \vec{v}}{\|\vec{u}\| \|\vec{v}\|},$$

**Step 2: Tangential Angles and Plane Reduction**

- **Tangential Angles**: Tangential angles describe the orientation of vectors within the specific 2D plane they span. These are defined relative to a chosen reference axis in that plane.

- **Plane Spanned by $\vec{u}$ and $\vec{v}$**: Any two vectors in $n$-dimensional space span a **2D subspace** (a plane). This means the interaction between $\vec{u}$ and $\vec{v}$ (e.g., the angle $\theta$) is fully determined by their projections into this plane.

- **Orthonormal Basis for the Plane**: Using the Gram-Schmidt process, construct an orthonormal basis $\{\vec{e}_1, \vec{e}_2\}$:

  - Normalize $\vec{u}$ to define $\vec{e}_1$:

$$\vec{e}_1 = \frac{\vec{u}}{\|\vec{u}\|}.$$

  - Define $\vec{e}_2$ as orthogonal to $\vec{e}_1$ and lying in the same plane:

$$\vec{e}_2 = \frac{\vec{v} - (\vec{v} \cdot \vec{e}_1)\vec{e}_1}{\|\vec{v} - (\vec{v} \cdot \vec{e}_1)\vec{e}_1\|}.$$

**Step 3: Expressing $\vec{u}$ and $\vec{v}$ in the Orthonormal Basis**

In the orthonormal basis $\{\vec{e}_1, \vec{e}_2\}$:

$$\vec{u} = \|\vec{u}\|\vec{e}_1,$$

and:

$$\vec{v} = a\vec{e}_1 + b\vec{e}_2,$$

where:

$$a = \vec{v} \cdot \vec{e}_1, \quad b = \vec{v} \cdot \vec{e}_2.$$

**Step 4: Computing $\cos\theta$**

The angle $\theta$ between $\vec{u}$ and $\vec{v}$ is:

$$\cos\theta = \frac{\vec{u} \cdot \vec{v}}{\|\vec{u}\|\|\vec{v}\|}.$$

Substituting:

$$\vec{u} \cdot \vec{v} = \|\vec{u}\|a, \quad \|\vec{v}\| = \sqrt{a^2 + b^2}.$$

Thus:

$$\cos\theta = \frac{a}{\sqrt{a^2 + b^2}}.$$

**Step 5: Computing $\sin\theta$**

The magnitude of the cross product $\|\vec{u} \times \vec{v}\|$ in the 2D plane is related to $\sin\theta$ by:

$$\|\vec{u} \times \vec{v}\| = \|\vec{u}\|\|\vec{v}\|\sin\theta.$$

Substituting:

$$\|\vec{u} \times \vec{v}\| = \|\vec{u}\||b|.$$

Thus:

$$\sin\theta = \frac{|b|}{\sqrt{a^2 + b^2}}.$$

**Step 6: Computing $\tan\theta$**

The tangent of $\theta$ is:

$$\tan\theta = \frac{\sin\theta}{\cos\theta}.$$

Substituting:

$$\tan\theta = \frac{\frac{|b|}{\sqrt{a^2+b^2}}}{\frac{a}{\sqrt{a^2+b^2}}}.$$

Simplify:

$$\tan\theta = \frac{|b|}{a}.$$

Thus:

$$\theta = |\arctan(b/a)|.$$

**Step 7: Relating $\theta$ to the Tangential Angles**

In the 2D plane:

- The tangential angle of $\vec{u}$ relative to $\vec{e}_1$ is:

$$\alpha_{\vec{u}} = 0 \quad (\vec{u} \text{ lies entirely along } \vec{e}_1).$$

- The tangential angle of $\vec{v}$ is:

$$\alpha_{\vec{v}} = \arctan(b/a).$$

The difference in tangential angles is:

$$|\alpha_{\vec{u}} - \alpha_{\vec{v}}| = |\arctan(b/a)|.$$

Thus, the geometric angle $\theta$ between $\vec{u}$ and $\vec{v}$ satisfies:

$$\theta = |\alpha_{\vec{u}} - \alpha_{\vec{v}}|.$$

This completes the proof. $\square$

Now we investigate the relationship between word embedding vector norms and the change of arc length. We start with the goal of approximating the arc length $\Delta s$ between two consecutive word embeddings, $\vec{u}$ and $\vec{v}$, in a high-dimensional space. Arc length is classically defined as the integral of the norm of the tangent vector along the curve. For discrete data points, this is approximated as a sum of the Euclidean distances between points.

**1. Discrete Approximation of Arc Length**

Given consecutive embeddings $\vec{u}$ and $\vec{v}$, the arc length between these points can be approximated as:

$$\Delta s = \|\vec{u} - \vec{v}\|.$$

However, directly using $\|\vec{u} - \vec{v}\|$ would treat the embeddings purely as geometric points and ignore their semantic significance as encoded by the vector magnitudes.

**2. Semantic Weight and Embedding Norms**

In NLP, the norm of a word embedding, $\|\vec{u}\|$, encodes the semantic "weight" or importance of a word within its context Schakel & Wilson (2015); Reif et al. (2019). Larger norms indicate that the embedding carries more semantic information, while smaller norms suggest less significance.

For two consecutive embeddings, $\vec{u}$ and $\vec{v}$, their combined semantic importance is proportional to their norms:

$$\text{Semantic Importance} \propto \|\vec{u}\| + \|\vec{v}\|.$$

**3. Inverse Proportionality and Arc Length**

To align with the geometric principle in Whewell's equation that relates curvature ($\kappa$) and arc length ($\Delta s$) as:

$$\kappa \propto \frac{1}{\Delta s},$$

we posit that arc length ($\Delta s$) should *decrease* when the semantic importance ($\|\vec{u}\| + \|\vec{v}\|$) increases.

This motivates the choice of the *inverse relationship*:

$$\Delta s \propto \frac{1}{\|\vec{u}\| + \|\vec{v}\|}.$$

**4. Sum of Inverse Norms as Arc Length**

While $\|\vec{u}\| + \|\vec{v}\|$ represents the combined semantic importance of two embeddings, directly using it in the denominator would contradict the inverse proportionality between $\Delta s$ and $\kappa$. Instead, we take the *inverse of the norms individually*, which ensures the arc length is smaller for larger semantic weights.

Thus, the arc length approximation becomes:

$$\Delta s \propto \frac{1}{\|\vec{u}\|} + \frac{1}{\|\vec{v}\|}.$$

The reasoning is that embeddings with larger norms (higher semantic significance) should have smaller contributions to the overall arc length, reflecting the sharper semantic transitions between significant words.

### A.1.3 TIME AND SPACE COMPLEXITY OF CURVALID

The time complexity for PromptLID is $\mathcal{O}(np)$, where $n$ is the number of prompts and $p$ is the dimensionality of the prompt embeddings. In our implementation, we use the representation layer of the CNN to obtain the embeddings. For TextCurv, the time complexity is $\mathcal{O}(nmd)$, where m is the number of words in a prompt and d is the word embedding dimensionality. In our case, we use RoBERTa embeddings for word representations. Therefore, the overall time complexity is $\mathcal{O}(n(p + md))$.

The space complexity for PromptLID is $\mathcal{O}(np)$, as we store n prompts, each with $p$ dimensions. For TextCurv, the space complexity is $\mathcal{O}(nz)$, where $z$ represents the dimensionality of the trained CNN layers used in our computations. Consequently, the space complexity is $\mathcal{O}(n(p + z))$.

### A.1.4 CNN HYPERPARAMETER SELECTION

We conducted a preliminary study for the CNN hyperparameter selection to determine the optimal architecture based on overall CurvaLID detection accuracy, training time, and stability. Stability was evaluated by measuring the average CurvaLID accuracy across 10 random seeds. The study involved experimenting with the following parameters:

**Number of convolutional layers:** Tested configurations with 1 to 5 layers.
**Activation functions:** Evaluated ReLU, ELU, tanh, sigmoid, and softplus.
**Kernel sizes:** Tested kernel sizes ranging from 2 to 5.
**Number of parameters:** Adjusted the dense layer sizes to 64, 128, and 256 units.
**Epochs:** Tested training with 10, 20, 30, 40, 50 epochs.
**Batch sizes:** Evaluated sizes of 16, 32, 64, and 128.
**Optimizers:** Compared Adam and SGD.

The experiment result is shown in Table 5.

### A.1.5 DETAILED PARAMETERS AND SPECIFICS OF THE CNN ARCHITECTURE FOR CLASSIFYING BENIGN PROMPTS IN CURVALID STEP 1

The CNN architecture consists of an input layer and two 1D convolutional layers. The first Conv1D layer applies 32 filters with a kernel size of 3 and a ReLU activation function, while the second Conv1D layer increases the number of filters to 64, again using a kernel size of 3 and ReLU activation. The output from the convolutional layers is flattened before passing through a fully connected layer with 128 units and ReLU activation. Finally, the network includes an output layer with four units and a softmax activation to classify the input into four distinct categories: Orca, MMLU, AlphEval, and TQA. The model is compiled using the Adam optimizer, categorical cross-entropy loss, and accuracy as the evaluation metric. Training is conducted over 20 epochs with a batch size of 32 and a validation split of 20%.

### A.1.6 DETAILED PARAMETERS AND SPECIFICS OF THE MLP ARCHITECTURE FOR CLASSIFYING BENIGN AND ADVERSARIAL PROMPTS IN CURVALID STEP 4

The MLP architecture consists of two fully connected layers and an output layer. The first layer contains 256 neurons with ReLU activation, followed by a batch normalization and dropout layer with a rate of 0.5 to prevent overfitting. A second layer, with 128 neurons and ReLU activation, is followed by another batch normalization and dropout layer. The final output layer uses softmax activation with two units corresponding to the binary classification of benign and adversarial prompts. The

Table 5: Performance metrics for CNN hyperparameter selection.

| Hyperparameter | Overall Accuracy | Time (min) |
|---|---|---|
| **No. of Conv. Layers** | | |
| 1 | 0.962 | 13.2 |
| 2 | 0.992 | 14.6 |
| 3 | 0.992 | 14.6 |
| 4 | 0.974 | 15.8 |
| 5 | 0.993 | 15.7 |
| **Activation Function** | | |
| ReLU | 0.992 | 14.8 |
| ELU | 0.993 | 15.1 |
| tanh | 0.977 | 15.1 |
| Sigmoid | 0.975 | 14.5 |
| Softplus | 0.991 | 14.8 |
| **Kernel Size** | | |
| 2 | 0.942 | 14.8 |
| 3 | 0.992 | 14.9 |
| 4 | 0.988 | 14.8 |
| 5 | 0.982 | 15.0 |
| **Dense Layer Size** | | |
| 64 | 0.958 | 14.8 |
| 128 | 0.990 | 15.2 |
| 256 | 0.992 | 15.6 |
| **Epochs** | | |
| 10 | 0.943 | 14.2 |
| 20 | 0.991 | 14.9 |
| 30 | 0.989 | 15.3 |
| 40 | 0.990 | 15.5 |
| 50 | 0.988 | 15.3 |
| **Batch Size** | | |
| 16 | 0.990 | 15.6 |
| 32 | 0.991 | 14.8 |
| 64 | 0.988 | 14.7 |
| 128 | 0.982 | 14.2 |
| **Optimizer** | | |
| Adam | 0.990 | 15.2 |
| SGD | 0.981 | 15.8 |

model is compiled using the Adam optimizer, a learning rate of 0.001, and categorical cross-entropy as the loss function, and it is trained over 150 epochs with early stopping to prevent overfitting.

### A.1.7 EXPERIMENT SETTING FOR SECTION 5.1

We gathered 2519 prompts in total, with 1200 benign prompts and 1319 adversarial prompts. For benign prompts, we randomly sampled 300 prompts from each of Orca, MMLU, AlphacaEval and TruthfulQA. Besides, we gathered 320 SAP200 prompts, which is randomly selected 40 prompts from 8 adversarial goals. For DAN, we randomly sampled 350 prompts from the adversarial prompts uploaded on their Github, roughly half of their total number of prompts. For MathAttack, we used all 300 adversarial prompts uploaded to their Github. For GCG, we used their default parameter

setting: learning rate = 0.01, batch size = 512, top-k = 256, and temperature = 1, and created a universal adversarial suffix. The targetted model is Vicuna-7B-v1.5. We used all 349 adversarial behaviors as listed in their Github.

## A.2 ABLATION STUDIES

This section presents ablation studies for CurvaLID, including its performance across various adversarial prompts and its effectiveness under different training conditions and parameter settings.

### A.2.1 PERFORMANCE METRICS FOR CURVALID IN RANDOMSEARCH, AMPLEGCG, PERSUASIVE ATTACK, AUTODAN, AND DRATTACK

Table 6 shows the Performance metrics for CurvaLID in PAIR, RandomSearch, AmpleGCG, Persuassive Attack, AutoDAN, and DrAttack. Note that due to the abundance of each dataset, we are testing these adversarial datasets against benign datasets individually, i.e., four benign datasets against each adversarial dataset.

We obtained 100 data from each of the four benign datasets. For PAIR, we sampled 171 adversarial prompts by generating PAIR adversarial prompts targeting Palm2, GPT4, Vicuna-7B, and Llama-2-13b-chat-hf. This is because PAIR's code focuses on 50 adversarial goals for each LLM, and PAIR does not always find an adversarial prompt for each goal in its default setting. For Random-Search, we obtain the adversarial prompts from their GitHub directly. We gathered the datasets from attacking vicuna-13B-v1.5, R2D2, Nemotron-4-340b, Llama-2-70b, Llama-2-13B, GPT4_Turbo, GPT3.5_Turbo, and Gemma-7B. We randomly selected 200 unique adversarial prompts, as we observed that some adversarial prompts are the same when attacking different LLMs. For AmpleGCG, we obtained permission to access their adversarial prompts directly. We randomly sampled 200 prompts from the attack to Vicuna-7B on AdvBench. For Persuasive Attack, we included 150 adversarial prompts uploaded by the authors on HuggingFace. For AutoDAN, we included 150 Auto-DAN prompts generated by following the setting from their Github, with 50 prompts each targeting Vicuna-7b-v1.5, Llama2-7b, and GPT-4-0613. For DrAttack, we tested 150 adversarial prompts generated by following the setting from their Github, with 50 prompts targeting each of Vicuna-7B, LLaMA-2-7B-Chat, and GPT-3.5-turbo.

Table 6: Performance metrics for CurvaLID in PAIR, RandomSearch, AmpleGCG, Persuasive Attack, AutoDAN, and DrAttack.

| Adv. Dataset | PAIR | RandomSearch | AmpleGCG | Persuasive Attack | AutoDAN | DrAttack | Avg. |
|---|---|---|---|---|---|---|---|
| **Benign Acc.** | 0.973 | 1 | 0.975 | 0.952 | 0.973 | 0.975 | 0.975 |
| **Adv. Acc.** | 1 | 1 | 0.976 | 1 | 1 | 1 | 0.996 |
| **Overall Acc.** | 0.983 | 1 | 0.975 | 0.962 | 0.978 | 0.980 | 0.986 |
| **F1 Score** | 0.986 | 1 | 0.987 | 0.974 | 0.986 | 0.987 | 0.985 |

### A.2.2 ACCURACY OF CURVALID WITH LESS DATA

Table 7 shows the performance of CurvaLID when trained with less data. We training and tested CurvaLID with 150 data from each dataset, halving the number of data used from the main result. All other parameters remained the same.

### A.2.3 PERFORMANCE METRICS FOR CURVALID WITH REPLACING MLP BY LOCAL OUTLIER FACTOR OR ISOLATION FOREST

The parameters of the local outlier factor is as follows:n_neighbors=30, metric='chebyshev', leaf_size=10, and p=1.

For isolation forest, the contamination is set as auto.

We are testing 1900 prompts in total, with 100 prompts randomly sampled from each of SAP, DAN, MathAttack, GCG, PAIR, AmpleGCG and RandomSearch, and 300 prompts from each of Orca, MMLU, AlphacaEval and TruthfulQA. The experimental results are shown in Tables 8 and 9

Table 7: Classification Accuracy and Performance Metrics for CurvaLID on Benign and Adversarial Datasets with 150 Data from Each Dataset.

| Data Class | Accuracy by Dataset | | | | Accuracy by Class | Accuracy | F1 |
|---|---|---|---|---|---|---|---|
| **Benign** | **Orca** | **MMLU** | **AlpacaEval** | **TQA** | **0.992** | **0.988** | **0.988** |
| | 0.9565 | 1.000 | 1.000 | 1.000 | | | |
| **Adversarial** | **SAP** | **DAN** | **MathAttack** | **GCG** | **0.983** | | |
| | 1.000 | 0.966 | 1.000 | 0.969 | | | |

Table 8: Performance metrics for CurvaLID with replacing MLP by local outlier factor

| Metric | Benign | Adversarial | Overall |
|---|---|---|---|
| **Accuracy** | 0.953 | 0.840 | 0.909 |
| **F1 Score** | 0.927 | 0.878 | 0.903 |

Table 9: Performance metrics for CurvaLID with replacing MLP by isolation forest

| Metric | Benign | Adversarial | Overall |
|---|---|---|---|
| **Accuracy** | 0.910 | 0.902 | 0.907 |
| **F1 Score** | 0.953 | 0.833 | 0.903 |

Table 10: Ablation study comparing LID and TextCurv for benign and adversarial prompt classification.

| | **PromptLID** | **TextCurv (1st Conv. Layer)** | **TextCurv (2nd Conv. Layer)** | **TextCurv (Both Conv. Layers)** |
|---|---|---|---|---|
| **Benign Acc.** | 0.987 | 0.690 | 0.738 | 0.960 |
| **Adv. Acc.** | 0.932 | 0.833 | 0.809 | 0.884 |
| **Overall Acc.** | 0.958 | 0.783 | 0.783 | 0.777 |
| **F1 Score** | 0.958 | 0.781 | 0.782 | 0.776 |

### A.2.4 CURVALID WITH PROMPTLID OR TEXTCURV ONLY

We demonstrate that both PromptLID and TextCurv are essential for achieving optimal performance in CurvaLID. When using only PromptLID as the input feature, the model reaches 0.95 accuracy. However, by combining PromptLID and TextCurv, we achieve an accuracy of over 0.99.

Table 10 illustrated the performance of CurvaLID if we only use LID or TextCurv as our features.

### A.2.5 ACCURACY OF CURVALID IN DIFFERENT EMBEDDING

We tested CurvaLID using different word embeddings, including popular ones like GPT-2, BERT, XLNet, and DistilBERT. The experimental results show that CurvaLID performs similarly with around 0.99 overall accuracy, regardless of the word embedding used. Therefore, CurvaLID's classification performance is independent of the word embedding used.

Table 11 shows the accuracy of CurvaLID in different embedding. The experimental result shows that CurvaLID maintains a high classification accuracy under different word embeddings.

Table 11: Performance of CurvaLID with Different Word Embeddings. The table summarizes the classification accuracy and F1 scores for benign and adversarial prompts using various word embeddings.

| Word embedding | Benign Prompt Accuracy | Adv. Prompt Accuracy | Overall Accuracy | F1 |
|---|---|---|---|---|
| RoBERTa | 0.984 | 1.000 | 0.992 | 0.992 |
| GPT-2 | 0.973 | 1.000 | 0.986 | 0.986 |
| BERT | 0.987 | 1.000 | 0.994 | 0.993 |
| XLNet | 0.991 | 0.989 | 0.990 | 0.990 |
| DistilBERT | 0.970 | 0.992 | 0.982 | 0.980 |

Table 12: Attack success rates (ASR) in percentage after CurvaLID in vicuna-7b-v1.5.

| | SAP | DAN | MathAttack | GCG | PAIR | RandomSearch | AmpleGCG |
|---|---|---|---|---|---|---|---|
| **Vanilla** | 69 | 41 | 56 | 95 | 98 | 95 | 97.5 |
| **CurvaLID** | 0 | 0 | 0 | 0 | 0 | 0 | 2.4 |

Table 13: Comparison of CurvaLID Architectures

| Metric | CNN (Original Setting) | Transformer | RNN |
|---|---|---|---|
| Detection accuracy | 0.992 | 0.984 | 0.989 |
| Overall training time (min) | 14.58 | 39.01 | 25.10 |

### A.2.6 CurvaLID in defending Vicuna-7B-v1.5

We measured the reduction in ASR after CurvaLID identified and filtered out the adversarial attacks in Vicuna-7B-v1.5, using the same settings specified in the respective original adversarial prompt papers. As shown in Table 12, CurvaLID successfully reduced the ASR of most attacks to zero, outperforming the studied SOTA defenses.

### A.2.7 Replacement of CNN in CurvaLID Step 1 with Transformer and RNN models

We experimented with replacing the CNN architecture (Step 1 of CurvaLID) with Transformer and RNN models. The experimental results are shown in Table 13. While both Transformer and RNN achieved comparable detection accuracy (0.98 versus CNN's 0.992), they required almost two to three times longer training times. Details of the Transformer and RNN configurations are provided below:

**Transformer** The transformer has an input layer, a multi-head attention layer with 4 heads and a key dimension of 64, followed by layer normalization, a dense layer with 128 units, dropout (rate of 0.1), and a final layer normalization. The model then flattens the output, adds another dense layer with 128 units, and concludes with a softmax output layer for classification into 4 classes.

**RNN:** The RNN model begins with an input layer, followed by two stacked LSTM layers with 64 units each (the first LSTM layer returns sequences, while the second does not). After the LSTM layers, there is a dense layer with 128 units and a ReLU activation, followed by a softmax output layer for classification into 4 classes. The model is compiled with the Adam optimizer and categorical cross-entropy loss.

### A.2.8 Performance of CurvaLID on adversarial prompts with reordered word sequences

We utilized GPT-4-o to reorder the words in every sentence of the adversarial prompts while preserving their semantic meaning. The experimental results, presented in Table 14, demonstrate that CurvaLID maintains robust performance, achieving an overall accuracy of 0.984 in detecting these adversarial prompts with altered word order. It is important to note, however, that reordering the

Table 14: Performance metrics for CurvaLID on adversarial prompts with reordered word sequences.

| Data class | Accuracy by dataset | | | | Accuracy by class | Overall accuracy | F1 score |
|---|---|---|---|---|---|---|---|
| **Benign** | **Orca** | **MMLU** | **AlpacaEval** | **TQA** | **0.981** | | |
| | 0.993 | 0.983 | 0.973 | 0.973 | | **0.984** | **0.984** |
| **Adversarial** | **SAP** | **DAN** | **MathAttack** | **GCG** | **0.987** | | |
| | 0.983 | 0.963 | 1.000 | 1.000 | | | |

Table 15: Performance of CurvaLID on reordered social-engineered attacks

| Adversarial attack | PAIR | DAN | Persuasive Attack |
|---|---|---|---|
| **Benign accuracy** | 0.951 | 0.971 | 0.966 |
| **Adversarial accuracy** | 0.988 | 1.000 | 0.962 |
| **Overall accuracy** | 0.962 | 0.983 | 0.964 |
| **Attack success rate on Vicuna-7B-v1.5** | 0 | 0 | 0 |

words in adversarial prompts may potentially disrupt their effectiveness as attacks, as the content and intent of the original prompts could be compromised.

### A.2.9 PERFORMANCE OF CURVALID ON REORDERED PERSUASIVE SOCIAL-ENGINEERED ADVERSARIAL PROMPTS

In this section, we evaluate CurvaLID on PAIR, DAN, and Persuasive attacks, all of which are social-engineered persuasive attacks designed to preserve both semantic meaning and adversarial intent while varying structure Chao et al. (2023); Shen et al. (2023); Zeng et al. (2024). To introduce linguistic variations, we utilized GPT-4-o to reorder the words in each sentence of the adversarial prompts while maintaining their semantic meaning. The experimental setup remains the same as described in A.2.1, with the addition of 300 DAN prompts. The experimental results, presented in Table 15, show that CurvaLID consistently achieved over 96% detection accuracy across all three attack types and maintained a 0% attack success rate on Vicuna. These findings highlight the robustness of our method, even against sophisticated and linguistically varied prompts specifically crafted to bypass defenses.

### A.2.10 DIFFERENCES IN PROMPTLID AND TEXTCURV BETWEEN BENIGN AND ADVERSARIAL PROMPTS UNDER LINGUISTIC REORDERING

We conducted an experiment to investigate the differences in PromptLID and TextCurv between benign and adversarial prompts after reordering the adversarial prompts. To introduce linguistic variations, we utilized GPT-4-o to reorder the words in each sentence of the adversarial prompts while preserving their semantic meaning. For the benign prompts, we tested 100 samples each from Orca, MMLU, AlpacaEval, and TQA datasets. Similarly, for the adversarial prompts, we tested 100 samples each from PAIR, DAN, and Persuasive attacks Chao et al. (2023); Shen et al. (2023); Zeng et al. (2024). Table 16 highlights the geometric differences between the benign and adversarial prompts, demonstrating the effectiveness of our method in capturing these variations.

### A.2.11 CURVALID WITH SEPARATED BENIGN TRAINING AND TESTING DATA

We conducted an ablation study by training CurvaLID on two benign datasets and testing it on the remaining two. Specifically, we trained CurvaLID using only Orca and MMLU as benign data and evaluated it on AlpacaEval and TQA. The results, shown in Table 17, demonstrate an overall detection accuracy of 0.982, just one percentage point lower than when trained on all four benign datasets. These findings indicate that CurvaLID's performance remains robust and is not overly optimistic, even when tested on unseen benign datasets.

### A.2.12 CURVALID WITH LONG TEXT LENGTH BENIGN PROMPTS

We conducted an additional experiment to evaluate CurvaLID's performance on benign prompts with longer text lengths. Specifically, we calculated the median text length of benign prompts (106

Table 16: Geometric differences in TextCurv and PromptLID between benign and adversarial prompts. The percentages in parentheses indicate the relative increase in adversarial prompts compared to benign prompts, calculated as $(\text{Adversarial} - \text{Benign})/\text{Benign} \times 100$.

| Geometric Measures | TextCurv@Conv Layer 1 | | TextCurv@Conv Layer 2 | | PromptLID@Dense Layer | |
|---|---|---|---|---|---|---|
| | Benign | Adversarial | Benign | Adversarial | Benign | Adversarial |
| Average Value | 0.644 | 0.813 (+26.3%) | 0.341 | 0.425 (+24.6%) | 3.546 | 18.223 (+413.8%) |

Table 17: CurvaLID with different training and testing benign datasets

| Data class | Accuracy by dataset | | | | Accuracy by class | Overall accuracy | F1 score |
|---|---|---|---|---|---|---|---|
| Benign | AlpacaEval | | TQA | | 0.9412 | | |
| | 0.963 | | 0.9194 | | | 0.982 | 0.98 |
| Adversarial | SAP | DAN | MathAttack | GCG | 1 | | |
| | 1 | 1 | 1 | 1 | | | |

characters in our experiment), removed all benign prompts with fewer than the median, and reevaluated CurvaLID's performance as outlined in Section 5.1 (Page 7, Line 351). The results, presented in Table 18 below, show that this adjustment had minimal effect on detection accuracy. The overall accuracy was 0.990, compared to 0.992 when all benign prompts (without filtering by text length) were included. This confirms that text length does not materially impact CurvaLID's performance.

### A.2.13 CurvaLID with non-standard benign prompts

We conducted an experiment to evaluate CurvaLID's performance on non-standard benign samples. Specifically, we utilized GPT-4-o to introduce spelling errors by replacing one word in each sentence of all benign prompts with a misspelled variant. The experimental results are presented in Table 19 below.

Our findings reveal that the detection accuracy by dataset exhibited minimal changes, and the overall accuracy remained almost identical to the original experiment , which involved benign prompts without spelling errors. These results demonstrate that introducing spelling errors has a negligible impact on CurvaLID's performance, reaffirming its robustness in handling non-standard text inputs.

### A.3 LID analysis

This section includes supplementary information on LID analysis.

### A.3.1 LID Estimation using Method of Moments

This section provides the pseudo code for estimating LID by the Method of Moments, see Algorithm A.3.1.

---

**Algorithm A.2** LID Estimation using Method of Moments

---

**Require:** Dataset, Reference points, Number of neighbors $k$
1: **For each data point in Dataset:**
2:     Compute pairwise distances $r$ between the data point and all points in Reference
3:     Sort distances in ascending order and store them as $a$
4:     Compute the mean of the first $k-1$ nearest distances: $m = \frac{1}{k-1}\sum_{i=1}^{k-1} a_i$
5:     Estimate LID for the data point: $\text{LID} = \frac{m}{a_k - m}$
6: **Return** LID for all data points

---

### A.3.2 Average LID and standard deviation of prompts with and without stop words and punctuation

Table 20 shows the average LID and standard deviation of prompts with and without stop words and punctuation.

Table 18: CurvaLID on benign prompts over 106 characters

| Data class | Accuracy by dataset | | | | Accuracy by class | Overall accuracy | F1 score |
|---|---|---|---|---|---|---|---|
| Benign | Orca | MMLU | AlpacaEval | TQA | 0.981 | | |
| | 0.922 | 1.000 | 1.000 | 1.000 | | 0.990 | 0.990 |
| Adversarial | SAP | DAN | MathAttack | GCG | 1.000 | | |
| | 1.000 | 1.000 | 1.000 | 1.000 | | | |

Table 19: CurvaLID on benign prompts with spelling errors

| Data class | Accuracy by dataset | | | | Accuracy by class | Overall accuracy | F1 score |
|---|---|---|---|---|---|---|---|
| Benign | Orca | MMLU | AlpacaEval | TQA | 0.985 | | |
| | 0.957 | 0.983 | 1.000 | 1.000 | | 0.990 | 0.990 |
| Adversarial | SAP | DAN | MathAttack | GCG | 0.995 | | |
| | 0.990 | 0.990 | 1.000 | 1.000 | | | |

### A.3.3 TOP 10 MOST COMMON NEAREST NEIGHBOR WORDS AFTER REMOVING STOP WORDS AND PUNCTUATION

Table 21 shows the top 10 most common Nearest Neighbor words after removing stop words and punctuation. The experimental result demonstrates that word-level LID is insufficient for distinguishing between benign and adversarial prompts, even after the removal of stop words and punctuation.

### A.4 PERFORMANCE OF OTHER SOTA DEFENSES

We summarize the performance of various defenses across different LLMs, focusing on their ability to reduce the ASR of adversarial prompts and compare this to CurvaLID's unified performance. CurvaLID outperforms the studied defenses and maintains consistent performance across all LLMs. While we primarily compare four key defenses in this subsection—SmoothLLM (Robey et al., 2023), IntentionAnalysis (Zhang et al., 2024a), RTT3d(Yung et al., 2024), and LAT(Sheshadri et al., 2024)—which are considered SOTA or represent some of the most recent developments, we also experimented with other defenses like Gradient Cuff(Hu et al., 2024), SELFDEFEND(Wang et al., 2024), and SafeDecoding(Xu et al., 2024a).

Given the computational and time constraints associated with replicating results and testing across multiple LLMs, we have cited the performance figures for these defenses from their respective papers. This approach ensures fairness, as different studies report varying results for these defenses when replicating them against different adversarial prompts and across different models. Therefore, we rely on the original reported performances to provide a balanced and consistent comparison. Note that since the results for these defenses are primarily based on English adversarial prompts in their respective papers, our analysis here is focused solely on English prompts.

Table 20: Comparison of Average LID and Standard Deviation (SD) of Prompts with and without Stop Words and Punctuation

| Data Type | Dataset | With Stop Words and Punctuation | | Without Stop Words and Punctuation | |
|---|---|---|---|---|---|
| | | Avg. LID | SD | Avg. LID | SD |
| Benign | Orca | 9.77 | 2.52 | 6.08 | 1.91 |
| | MMLU | 9.23 | 2.86 | 5.64 | 1.76 |
| | AlpacaEval | 13.14 | 104.58 | 4.07 | 2.26 |
| | TQA | 6.29 | 3.40 | 3.93 | 0.65 |
| Adversarial | SAP | 12.10 | 0.55 | 7.27 | 0.46 |
| | DAN | 24.52 | 244.72 | 7.18 | 1.18 |
| | MWP | 8.98 | 1.44 | 4.14 | 1.28 |
| | GCG | 8.35 | 0.40 | 6.18 | 0.41 |

Table 21: Top 10 Most Common Nearest Neighbor Words for each dataset after removing stop words and punctuation. The symbol ␣ represents a space preceding a word or punctuation.

| Dataset | Top 10 Most Common Nearest Neighbor Words after removing stop words and punctuation |
|---|---|
| SAP | ␣Remember, ␣write, ␣goal, ␣mission, ␣act, ␣suicide, ␣Use, ␣phrases, ␣use, ␣refer |
| DAN | PT, ␣D, G, AN, ␣Chat, ␣answer, ␣AI, ␣responses, ␣response, ␣respond |
| MathAttack | ␣many, ␣much, ␣would, ␣apples, ␣money, ␣20, ␣sold, ␣5, ␣bought, ␣day |
| GCG | use, ␣mar, ␣text, dt, end, ate, c, package, ized, ␣t |
| Orca | ␣answer, à, °, ␣question, à¦, ␣one, ␣following, ␣said, s, ␣Answer |
| MMLU | ␣mortgage, acre, ␣state, ␣contract, ␣deed, ␣would, ␣question, ␣statute, ␣action, s |
| AlpacaEval | ␣drinks, ␣gathering, ␣interested, ␣give, ␣time, ␣home, br, ␣trying, ␣dishes, ␣guests |
| TQA | ␣say, ks, ␣Oz, ␣established, ␣famous, ␣primed, ␣mirror, es, ␣principle, ␣power |

Defenses based on input perturbation demonstrate mixed results depending on the LLM and the nature of the adversarial attack. For instance, IntentionAnalysis reduces the ASR to between 0.03% and 8.34%, but it struggles with models like Vicuna-7B and MPT-30B-Chat, where the ASR for the SAP attack can reach nearly 20%. Similarly, while SmoothLLM can reduce the ASR to nearly 0% in various LLMs, it fails against PAIR attacks, showing ASRs of 46% in Vicuna-13B and 24% in GPT-4. RTT3d, as the first defense against MathAttack, managed to mitigate 40% of MathAttack in GPT4, but it failed to reduce the ASR to under 10%. LAT achieves near-zero ASR for models like Llama2-7B-Chat and Llama3-8B-Instruct. However, its reliance on a white-box setting and its testing on models with fewer than 10 billion parameters limit its broader applicability.

In the remainder of this subsection, we present the defensive performance of the SOTA defenses. The performance figures for all defenses are directly cited from their original papers due to the computational and time constraints involved in replicating results and testing across various LLMs. This approach ensures consistency and fairness in comparison, as the results reported by different studies often vary when replicating these defenses on different adversarial prompts and models. By relying on the figures from the original sources, we aim to provide an accurate and balanced reflection of each defense's performance.

We first compared the defenses using a common LLM, LLaMA-2-7B-Chat, as all the studied defenses employ this model. Table 22 presents the ASR comparison between CurvaLID and seven other defenses, showcasing CurvaLID's exceptional performance in mitigating various adversarial prompts.

Table 22: ASR comparison of CurvaLID with other defenses against different adversarial prompts on LLaMA-2-7B-Chat. A dash (-) indicates that experiments were not conducted for this setting in the original paper.

| LLM | defense | Adversarial Prompt | | | | | | |
|---|---|---|---|---|---|---|---|---|
| | | GCG | PAIR | DAN | AmpleGCG | SAP | MathAttack | RandomSearch |
| LLaMA-2-7B-Chat | SmoothLLM | 0.1 | 8 | - | 0 | - | - | 0 |
| | LAT | 0.007 | 0.025 | - | - | - | - | - |
| | Gradient Cuff | 0.012 | 0.23 | - | - | - | - | - |
| | IntentionAnalysis | 0 | - | 0.13 | - | 0 | - | - |
| | SELFDEFEND | 0.09 | 0.21 | 0.242 | - | - | - | - |
| | RTT3d | 0.17 | 0.043 | - | - | - | - | - |
| | SafeDecoding | 0 | 0.04 | - | - | - | - | - |
| | **CurvaLID (ours)** | 0 | 0 | 0 | 0.025 | 0 | 0 | 0 |

The following are the experimental settings and performances of the seven defenses we studied. Note that it includes various LLMs, namely Vicuna (Chiang et al., 2023), Llama-2(Touvron et al., 2023), Llama-3(AI@Meta, 2024), GPT-3.5(Brown, 2020), GPT-4(Achiam et al., 2023), (Anil et al., 2023), Claude-1(Anthropic, 2023a), Claude-2(Anthropic, 2023b), ChatGLM-6B(Zeng et al., 2022), MPT-30B-Chat(Team, 2023), DeepSeek-67B-Chatand(Bi et al., 2024). It also includes various adversarial prompts, namely, GCG(Zou et al., 2023), PAIR(Chao et al., 2023), DAN(Shen et al., 2023), AmpleGCG(Liao & Sun, 2024), SAP(Deng et al., 2023a), MathAttack(Zhou et al., 2024b), RandomSearch(Andriushchenko et al., 2024), Prefill(Haizelabs, 2023), Many-Shot(Anil et al., 2024), AutoDAN(Liu et al., 2023), (Mehrotra et al., 2023), (Wei et al., 2024), LRL(Yong et al., 2023), DrAttack(Li et al., 2024), (Chang et al., 2024), (Deng et al., 2023b), DeepInception(Li et al., 2023a), and Template(Yu et al., 2023).

**SmoothLLM** Detailed experimental settings are referred to (Robey et al., 2023). The experimental results are shown in Table 23.

Table 23: ASR comparison of different LLMs against adversarial prompts under SmoothLLM defense. A dash (-) indicates that experiments were not conducted for this setting in the original paper.

| LLM | Adversarial Prompt | | | |
|---|---|---|---|---|
| | GCG | PAIR | RandomSearch | AmpleGCG |
| Vicuna-13B-v1.5 | 0.8 | 46 | 44 | 2 |
| Llama-2-7B-chat | 0.1 | 8 | 0 | 0 |
| GPT-3.5 | 0.8 | 2 | 0 | 0 |
| GPT-4 | 0.8 | 24 | 0 | 0 |
| PaLM-2 | 0.9 | - | - | - |
| Claude-1 | 0.3 | - | - | - |
| Claude-2 | 0.3 | - | - | - |

**Latent Adversarial Training** Detailed experimental settings are referred to (Sheshadri et al., 2024). The experimental results are shown in Table 24.

Table 24: ASR comparison of different LLMs against adversarial prompts under LAT defense.

| LLM | Adversarial Prompt | | | | |
|---|---|---|---|---|---|
| | PAIR | Prefill | AutoPrompt | GCG | Many-Shot |
| Llama2-7B-chat | 0.025 | 0.029 | 0.006 | 0.007 | 0 |
| Llama3-8B-instruct | 0.0033 | 0.0068 | 0 | 0.009 | 0 |

**Gradient Cuff** Detailed experimental settings are referred to (Hu et al., 2024). The experimental results are shown in Table 25.

Table 25: ASR comparison of different LLMs against adversarial prompts under Gradient Cuff defense.

| LLM | Adversarial Prompt | | | | | |
|---|---|---|---|---|---|---|
| | GCG | AutoDAN | PAIR | TAP | Base64 | LRL |
| Llama2-7B-chat | 0.012 | 0.158 | 0.23 | 0.05 | 0.198 | 0.054 |
| Vicuna-7B-v1.5 | 0.108 | 0.508 | 0.306 | 0.354 | 0 | 0.189 |

**IntentionAnalysis** Detailed experimental settings are referred to (Zhang et al., 2024a). The experimental results are shown in Table 26.

Table 26: ASR comparison of different LLMs against adversarial prompts under IntentionAnalysis defense. A dash (-) indicates that experiments were not conducted for this setting in the original paper.

| LLM | Adversarial Prompt | | | | |
|---|---|---|---|---|---|
| | DAN | SAP200 | DeepInception | GCG | AutoDAN |
| ChatGLM-6B | 5.48 | 6.12 | 0 | 1 | 2 |
| LLaMA2-7B-Chat | 0.13 | 0 | 0 | 0 | 0 |
| Vicuna-7B-v1.1 | 3.42 | 0.31 | 0 | 0 | 10.5 |
| Vicuna-13B-v1.1 | 0.94 | 1.12 | 0 | 0 | 3.5 |
| MPT-30B-Chat | 5.38 | 19.2 | 4.78 | 4 | - |
| DeepSeek-67B-Chat | 3.78 | 1.56 | 7.57 | 2 | - |
| GPT-3.5 | 0.64 | 0 | 0 | 0 | - |

**SELFDEFEND** Detailed experimental settings are referred to (Wang et al., 2024). The experimental results are shown in Table 27.

Table 27: ASR comparison of different LLMs against adversarial prompts under SELFDEFEND defense.

| LLM | Adversarial Prompt | | | | | | | |
|---|---|---|---|---|---|---|---|---|
| | DAN | GCG | AutoDAN | PAIR | TAP | DrAttack | Puzzler | MultiJail |
| GPT-3.5 | 0.007 | 0.18 | 0.31 | 0.29 | 0.02 | 0.71 | 0.22 | 0.203 |
| GPT-4 | 0.002 | 0 | 0.01 | 0.1 | 0.08 | 0.04 | 0.26 | 0.012 |

**RTT3d** Detailed experimental settings are referred to (Yung et al., 2024). The experimental results are shown in Table 28.

Table 28: ASR comparison of different LLMs against adversarial prompts under RTT3d defense. A dash (-) indicates that experiments were not conducted for this setting in the original paper.

| LLM | Adversarial Prompt | | | |
|---|---|---|---|---|
| | PAIR | GCG | SAP | MathAttack |
| GPT-3.5 | - | - | 0.06 | 9.8 |
| GPT-4 | 0.265 | - | - | - |
| Llama-2-13B-Chat | 0.043 | 0.17 | - | - |
| Vicuna-13B-v1.5 | 0.26 | 0.15 | - | - |
| PaLM-2 | 0.13 | - | - | - |

**SafeDecoding** Detailed experimental settings are referred to (Xu et al., 2024a). The experimental results are shown in Table 29.

Table 29: ASR comparison of different LLMs against adversarial prompts under SafeDecoding defense.

| LLM | Adversarial Prompt | | | | | |
|---|---|---|---|---|---|---|
| | GCG | AutoDAN | PAIR | DeepInception | SAP30 | Template |
| **Vicuna-7B** | 0.04 | 0 | 0.04 | 0 | 0.09 | 0.05 |
| **Llama2-7B-Chat** | 0 | 0 | 0.04 | 0 | 0 | 0 |

