# OpenReview forum: "CURVALID: A Geometrically-guided Adversarial Prompt Detection"
_ICLR.cc/2025/Conference — Submitted to ICLR 2025_

### Official Review · Reviewer_u7FH · 2024-11-03

**Soundness:** 3
**Presentation:** 2
**Contribution:** 2
**Rating:** 3
**Confidence:** 4

**Summary:**

This paper introduces CurvaLID, a model that detects adversarial prompts targeting LLMs. CurvaLID can capture the structural characteristics of entire prompts through "Prompt-Level Local Intrinsic Dimensionality" (PromptLID) and detects semantic shifts within the text using "Text Curvature" (TextCurv) to identify whether a prompt is adversarial. Since this method relies entirely on analyzing textual structure rather than LLM-specific architectures or parameters, it requires no fine-tuning for each LLM. This design significantly accelerates training speed, taking only 15 minutes on a single Nvidia H100 GPU.

**Strengths:**

The method has good advantages. Currently, most adversarial sample detection methods rely on large models, which makes the training process quite time-consuming. CurvaLID bypasses large language models entirely, directly analyzing text embeddings to identify adversarial inputs, and saving computational resources.

Additionally, the method is zero-shot, meaning it doesn't require fine-tuning for each LLM, making it even more versatile and efficient.

**Weaknesses:**

1 I think that the innovation and contributions of your method are somewhat limited. The proposed approach resembles a CNN-based text filter, primarily detecting anomalies in embeddings, which isn't a novel concept.

2 The connection between this work and LLMs is relatively weak. As mentioned, CurvaLID functions more like an independent text filter, focusing only on text structure and density features. It's more of a generic text filter than a targeted tool for detecting adversarial samples specific to LLMs. I don't mean to suggest that the detection method has no value, but most LLM-focused adversarial attacks today generate adversarial samples that closely resemble benign samples in terms of structure, density, and even embedding similarity. This would likely limit CurvaLID's effectiveness against subtle adversarial prompts.

3 There are also issues in the experimental design. CurvaLID was trained and evaluated on the same datasets—Orca, MMLU, AlpacaEval, and TQA—which could inevitably lead to overly optimistic results.

4 I noticed that the manuscript mixes American and British spellings in some places (such as "defence" and "defense"). To maintain a professional and consistent tone throughout, I recommend choosing one spelling style—American or British—and using it uniformly across the article.

5 The formatting of tables is inconsistent. Some tables use a three-line format, while others do not. A unified table style would improve clarity and presentation quality.

**Questions:**

1 Have the authors tried different CNN configurations to assess their impact on the performance of PromptLID and TextCurv?

2. Why did the authors choose to use CNN? If the goal is to capture relationships between textual features, wouldn’t Transformers perform better? Have you considered trying Transformers for this purpose?

3. The method relies only on structural and semantic changes to identify adversarial samples. Have the authors tested against adversarial examples with structural and semantic characteristics similar to benign samples? Specifically, have the authors tried generating adversarial examples with density and curvature that closely resemble benign samples to evaluate CurvaLID’s performance?

4. Have the authors analyzed the impact of text length on PromptLID? Given that longer texts have higher complexity and density, could they lead to benign texts being misclassified as adversarial?

5. Have the authors tested whether non-standard benign samples, such as those containing spelling errors or slang, might be misclassified as adversarial examples?

6. Can the method detect adversarial prompts that rely on contextual association across multiple prompts?

7. The authors trained the model on Orca, MMLU, AlpacaEval, and TQA datasets and then used the same datasets to evaluate whether the samples were benign or adversarial. Given this setup, does the model’s high accuracy have practical significance?

---

> ### Author Response · Authors · 2024-11-20
> **Author Response (Response to Weakness 1, 2, 3, 4, 5)**
>
> Thanks for your insightful reviews. Please find our response to your questions below:
>
> ---
>
> **Response to Weakness 1**
>
> We respectfully disagree with the reviewer's characterization of CurvaLID as merely a CNN-based text filter. A key part of our novelty lies in the insight that adversarial prompts can be effectively characterized by a geometric framework, specifically through curvature and local intrinsic dimensionality. The CNN feature extraction step is only one of four steps in the CurvaLID framework and serves an auxiliary purpose. As explicitly stated in our initial paper submission (e.g., Section 4.3, and Section 5.3), the CNN component is used to amplify the differences in TextCurv and PromptLID between benign and adversarial prompts. We would appreciate it if the reviewer could specifically point out the papers using CNN-based text filters similar to our approach. We are happy to provide further comparisons.
>
> We summarize the novel technical innovation, contributions, and novel findings that the reviewer might have missed.
>
> - To the best of our knowledge, this work is the first to use LID for adversarial text detection. We proposed the PromptLID, a prompt-level LID estimation that effectively captures the local dimensional differences between benign and adversarial prompts.
> - To the best of our knowledge, we are the first to investigate text curvature as a feature for adversarial prompt detection. We proposed TextCurv, a novel mathematical framework that captures the geometric properties of text prompts through curvature analysis, focusing on the degree of curvature in the manifolds.
> - CurvaLID is fundamentally different from existing SOTA LLM defenses in its model-agnostic design, efficiency, and exceptional performance.
> - We have revealed a novel phenomenon in the token-level LID estimation: the shared nearest neighbors being dominated by stop words and punctuation. Existing work using token-level LID for truthfulness evaluation did not consider such a phenomenon. Our novel phenomenon can benefit the community in future studies of token-level LID.
>
> ---
>
> **Response to Weakness 2**
>
> We have examined DAN and PAIR in which adversarial samples closely resemble benign samples in terms of structure, density, and even embedding similarity in our initial submission Table 1 (Page 8) and Table 6 (Page 19). We have also included Persuasive Attack, AutoDAN, and DrAttack which closely resemble benign prompts in our revised submission Table 6 (Page 22). Results show our detection is effective against these attacks. We would appreciate it if the reviewer could point out if there is any specific attack that we might have missed.
>
> ---
>
> **Response to Weakness 3**
>
> We would like to clarify that CurvaLID employs a train-test split strategy, using 80% of the data for training and 20% for testing. Importantly, there is minimal overlap among different benign datasets, and there is no overlap within individual datasets.
>
> To further address the concern, we conducted an ablation study by training CurvaLID on two benign datasets and testing it on the remaining two. Specifically, we trained CurvaLID using only Orca and MMLU as benign data and evaluated it on AlpacaEval and TQA. The results, shown in Table 17, demonstrate an overall detection accuracy of 0.982, just one percentage point lower than when trained on all four benign datasets. These findings indicate that CurvaLID's performance remains robust and is not overly optimistic, even when tested on unseen benign datasets. We thank the reviewer for their insightful comment and have incorporate the above discussion into the revised version of the paper, Table 17 in Appendix A.2.11, Page 26.
>
> | **Data class**    | **AlpacaEval** | **TQA**   |  |    |  | **Accuracy by class** | **Overall accuracy** | **F1 score** |
> |--------------------|----------------|-----------|---------|-------------------------|---------|------------------------|-----------------------|--------------|
> | **Benign**         | 0.963         | 0.9194    |         |         |                | **0.9412**       | **0.982**            |       **0.98**     |    |
> |                    |  **SAP**  |  **DAN** | **MathAttack** | **GCG**  |    |                        |                       |              |
> | **Adversarial**    |      1          |    1       | 1       | 1       |                    |    **1**              |                       |              |
>
> **Table 17:** CurvaLID with different training and testing benign datasets.
>
> ---
>
> **Response to Weakness 4**
>
> Thank you for your suggestion. We have updated the paper with consistent spelling.
>
> ---
>
> **Response to Weakness 5**
>
> Thank you for your suggestion. We have ensured that all tables adhere to a consistent style in the revised version of the paper.

---

> ### Author Response · Authors · 2024-11-20
> **Author Response (Response to Question 1)**
>
> **Response to Question1**
>
> We have provided an ablation study with a single convolutional layer or replacing the convolutional layers entirely with the dense layer in the initial submission in Appendix A.10 and Table 17. To address the reviewer's concerns, we added the following ablation study.
>
> We conducted a study for the CNN hyperparameter selection to determine the optimal architecture based on overall CurvaLID detection accuracy, training time, and stability. Stability was evaluated by measuring the average CurvaLID accuracy across 10 random seeds.  It is important to note that the reported overall accuracy refers specifically to CurvaLID's detection performance, and the running time reflects the total runtime of CurvaLID, not just the CNN component. Furthermore, for each experiment, the tested parameter was the only variable, while all other parameters and settings within CurvaLID remained consistent. The study involved experimenting with the following parameters:
>
> - **Number of convolutional layers:** Tested configurations with 1 to 5 layers.
> - **Activation functions:** Evaluated ReLU, ELU, tanh, sigmoid, and softplus.
> - **Kernel sizes:** Tested kernel sizes ranging from 2 to 5.
> - **Number of parameters:** Adjusted the dense layer sizes to 64, 128, and 256 units.
> - **Epochs:** Tested training with 10, 20, 30, 40,50 epochs.
> - **Batch sizes:** Evaluated sizes of 16, 32, 64, and 128.
> - **Optimizers:** Compared Adam and SGD.
>
> Results can be found in the table below and Table 5 (Appendix A.1.4) in the revised version. None of these hyperparameters have any significant impact on the detection performance.
>
> We would appreciate it if the reviewer could point out specifically what other hyperparameters should be tested.
>
> |  CNN hyperparameter  | Overall Accuracy | Time (min) |
> |:--------------------:|:----------------:|:----------:|
> |  No. of Conv. Layers |                  |            |
> |           1          |       0.962      |    13.2    |
> |           2          |       0.992      |    14.6    |
> |           3          |       0.992      |    14.6    |
> |           4          |       0.974      |    15.8    |
> |           5          |       0.993      |    15.7    |
> |  Activation function |                  |            |
> |         ReLU         |       0.992      |    14.8    |
> |          ELU         |       0.993      |    15.1    |
> |         tanh         |       0.977      |    15.1    |
> |        Sigmoid       |       0.975      |    14.5    |
> |       Softplus       |       0.991      |    14.8    |
> |      Kernel size     |                  |            |
> |           2          |       0.942      |    14.8    |
> |           3          |       0.992      |    14.9    |
> |           4          |       0.988      |    14.8    |
> |           5          |       0.982      |    15.0    |
> |   Dense layer size   |                  |            |
> |          64          |       0.958      |    14.8    |
> |          128         |       0.990      |    15.2    |
> |          256         |       0.992      |    15.6    |
> |         Epoch        |                  |            |
> |          10          |       0.943      |    14.2    |
> |          20          |       0.991      |    14.9    |
> |          30          |       0.989      |    15.3    |
> |          40          |       0.990      |    15.5    |
> |          50          |       0.988      |    15.3    |
> |      Batch size      |                  |            |
> |          16          |       0.990      |    15.6    |
> |          32          |       0.991      |    14.8    |
> |          64          |       0.988      |    14.7    |
> |          128         |       0.982      |    14.2    |
> |       Optimiser      |                  |            |
> |         Adam         |       0.990      |    15.2    |
> |          SGD         |       0.981      |    15.8    |
> Table 5: Performance metrics for CNN hyperparameter selection.

---

> ### Author Response · Authors · 2024-11-20
> **Author Response (Response to Question 2, 3)**
>
> **Response to Question 2**
>
> We selected CNN for its computational efficiency and high accuracy within our CurvaLID framework. The variable length sequences handled by a CNN are done by zero-padding. It only requires 15 minutes of training to achieve detection accuracy of 0.992.
>
> We experimented with replacing the CNN architecture with Transformer and RNN models. While both Transformer and RNN achieved comparable detection accuracy (0.98 versus CNN's 0.992), they required almost two to three times longer training times. Details of the Transformer and RNN configurations are provided in Remark [1], with the experimental results presented in Table 13:
>
>
> | Metric                      | CNN (Original Setting) | Transformer |  RNN  |
> |-----------------------------|:----------------------:|:-----------:|:-----:|
> | Detection Accuracy          |          0.992         |    0.984    | 0.989 |
> | Overall Training Time (min) |          14.58         |    39.01    | 25.10 |
>
> Table 13: Comparison of CurvaLID Architectures
>
>
> In conclusion, the CNN architecture remains the most suitable choice for our CurvaLID model, balancing computational efficiency with exceptional accuracy. We appreciate the reviewer's suggestion to test the Transformer and RNN and have included these experimental results in the revised version of our paper (Table 13 in Appendix A.2.7, Page 24).
>
> Remark [1]: The transformer we tested has an input layer, a multi-head attention layer with 4 heads and a key dimension of 64, followed by layer normalization, a dense layer with 128 units, dropout (rate of 0.1), and a final layer normalization. The model then flattens the output, adds another dense layer with 128 units, and concludes with a softmax output layer for classification into 4 classes. The RNN model begins with an input layer, followed by two stacked LSTM layers with 64 units each (the first LSTM layer returns sequences, while the second does not). After the LSTM layers, there is a dense layer with 128 units and a ReLU activation, followed by a softmax output layer for classification into 4 classes. The model is compiled with the Adam optimizer and categorical cross-entropy loss.
>
> ---
>
> **Response to Question 3**
>
> We tested CurvaLID against social-engineered and persuasive attacks, such as PAIR and DAN. These are specific types of subtle adversarial attacks written in grammatically correct, fluent English that are highly interpretable by humans. As shown in Table 1 (Page 8, Lines 378–384), Table 2 (Page 8, Lines 386–393), and Table 6 (Page 19, Lines 972–979) in our initial paper submission, CurvaLID demonstrates its effectiveness even against these challenging adversarial prompts, maintaining high detection accuracy.
>
> We attempted to generate adversarial prompts with density and curvature closely resembling benign samples by trial and error, guided by the adversarial goals outlined in AdvBench. We have included Table 28 below, detailing the techniques explored for generating adversarial prompts. These techniques include crafting persuasive arguments to request harmful content, creating imaginary scenarios, and appending socially-engineered adversarial queries with the GCG adversarial suffix. Each prompt was designed to closely resemble the curvature or LID of benign prompts. However, none of these attempts succeeded in successfully attacking the tested LLMs, including Vicuna-7B and LLaMA-2-7B-Chat. We would greatly appreciate any suggestions from the reviewer on how to design such an attack, and we are committed to following up on this valuable feedback.
>
> | Aim                                | Technique                                                          | Attack success rate |
> |------------------------------------|--------------------------------------------------------------------|---------------------|
> | Resemble benign prompt's Curvature | Prompt Engineering: Persuasion crafted through original writing    | 0                   |
> | Resemble benign prompt's Curvature | Prompt Engineering: Crafting imaginary scenarios and storytelling  | 0                   |
> | Resemble benign prompt's Curvature | Gibberish Suffix: Social-engineered adversarial query + GCG suffix | 0                   |
> | Resemble benign prompt's LID       | Prompt Engineering: Persuasion crafted through original writing    | 0                   |
> | Resemble benign prompt's LID       | Prompt Engineering: Crafting imaginary scenarios and storytelling  | 0                   |
> | Resemble benign prompt's LID       | Gibberish Suffix: Social-engineered adversarial query + GCG suffix | 0                   |
> Table 28: Techniques explored for generating adversarial prompts that resemble benign prompts’ curvature and LID.

---

> ### Author Response · Authors · 2024-11-20
> **Author Response (Response to Question 4, 5)**
>
> **Response to Question 4**
>
> We would like to clarify that text length has no theoretical or practical impact on PromptLID. Theoretically, in PromptLID, each prompt is represented as a single fixed-dimension vector. This process is detailed in Section 4.1 in our initial submission. Consequently, the text length does not directly influence the prompt representation. Furthermore, when calculating PromptLID, the local neighborhood of a prompt is defined by its nearest neighbors, as described in Section 4.1. Even if text length were to indirectly affect the prompt representation, its influence would be normalized within the local neighborhood during PromptLID computation.
>
> Practically, we conducted an additional experiment to evaluate CurvaLID's performance on benign prompts with longer text lengths. Specifically, we calculated the median text length of benign prompts (106 characters in our experiment), removed all benign prompts with fewer than the median, and reevaluated CurvaLID's performance as outlined in Section 5.1 (Page 7, Line 351). The results, presented in Table 18 below, show that this adjustment had minimal effect on detection accuracy. The overall accuracy was 0.990, compared to 0.992 when all benign prompts (without filtering by text length) were included. This confirms that text length does not materially impact CurvaLID's performance. We thank the reviewer for their insightful comment and incorporated the above discussion into the revised version of the paper (Appendix A.2.12, Table 18, Page 27).
>
> | **Data class**    | **Orca** | **MMLU**   |**AlpacaEval**  | **TQA**   |  | **Accuracy by class** | **Overall accuracy** | **F1 score** |
> |--------------------|----------------|-----------|---------|-------------------------|---------|------------------------|-----------------------|--------------|
> | **Benign**         | 0.922         | 1    |   1      |1        |                | **0.981**       | **0.990**            |  **0.990**         |     |
> |                    |  **SAP**  |  **DAN** | **MathAttack** | **GCG**  |    |                        |                       |              |
> | **Adversarial**    |      1          |    1       | 1       | 1       |                    |    **1**                |                       |              |
>
> **Table 18:** CurvaLID on benign prompts over 106 characters.
>
> ---
>
> **Response to Question 5**
>
> We conducted an experiment to evaluate CurvaLID's performance on non-standard benign samples. Specifically, we utilized GPT-4-o to introduce spelling errors by replacing one word in each sentence of all benign prompts with a misspelled variant. Subsequently, we performed the CurvaLID evaluation described in Section 5.1 (initial submission Page 7, Line 351). The experimental results are presented in Table 19 below.
>
> | **Data class**    | **Orca** | **MMLU**   |**AlpacaEval**  |**TQA**    |  | **Accuracy by class** | **Overall accuracy** | **F1 score** |
> |--------------------|----------------|-----------|---------|-------------------------|---------|------------------------|-----------------------|--------------|
> | **Benign**         | 0.957         | 0.983    |   1      |    1     |                | **0.985**       | **0.990**            |       **0.990**     |    |
> |                    |  **SAP**  |  **DAN** | **MathAttack** | **GCG**  |    |                        |                       |              |
> | **Adversarial**    |      0.99          |    0.99       | 1       | 1       |                    |    **0.955**              |                       |              |
> Table 19: CurvaLID on benign prompts with spelling errors
>
> Our findings reveal that the detection accuracy by dataset exhibited minimal changes, and the overall accuracy remained almost identical to the original experiment (Table 1, Page 8, Lines 378–386), which involved benign prompts without spelling errors. These results demonstrate that introducing spelling errors has a negligible impact on CurvaLID's performance, reaffirming its robustness in handling non-standard text inputs. We thank the reviewer for their insightful comment and have incorporated the above discussion into the revised version of the paper (Appendix A.2.13, Table 19, Page 26).

---

> ### Author Response · Authors · 2024-11-20
> **Author Response (Response to Question 6, 7)**
>
> **Response to Question 6**
>
> We would like to clarify that this paper focuses on single-step attacks, representing the primary research focus for many SOTA defenses against LLM adversarial attacks [1,2,3,7,8,9,10]. Other types of attacks often require fundamentally different experimental setups, such as few-shot configurations or additional pre-input instructions, which are beyond the scope of this study. We acknowledge the importance of other attack types, which we will explore in future works.
>
> ---
> **Response to Question 7**
>
> We would like to clarify that CurvaLID employs a train-test split strategy, using 80% of the data for training and 20% for testing. Importantly, there is minimal overlap among different benign datasets, and there is no overlap within individual datasets.
>
> To further address the concern, we conducted an ablation study by training CurvaLID on two benign datasets and testing it on the remaining two. Specifically, we trained CurvaLID using only Orca and MMLU as benign data and evaluated it on AlpacaEval and TQA. The results, shown in Table 17, demonstrate an overall detection accuracy of 0.982, just one percentage point lower than when trained on all four benign datasets. These findings indicate that CurvaLID's performance remains robust and is not overly optimistic, even when tested on unseen benign datasets. We thank the reviewer for their insightful comment and have incorporate the above discussion into the revised version of the paper, Table 17 in Appendix A.2.11, Page 26.
>
> | **Data class**    | **AlpacaEval** | **TQA**   |  |    |  | **Accuracy by class** | **Overall accuracy** | **F1 score** |
> |--------------------|----------------|-----------|---------|-------------------------|---------|------------------------|-----------------------|--------------|
> | **Benign**         | 0.963         | 0.9194    |         |         |                | **0.9412**       | **0.982**            |       **0.98**     |    |
> |                    |  **SAP**  |  **DAN** | **MathAttack** | **GCG**  |    |                        |                       |              |
> | **Adversarial**    |      1          |    1       | 1       | 1       |                    |    **1**              |                       |              |
>
> **Table 17:** CurvaLID with different training and testing benign datasets.
>
> ---
> [1]: Robey, A., Wong, E., Hassani, H., & Pappas, G. J. (2023). Smoothllm: Defending large language models against jailbreaking attacks. arXiv preprint arXiv:2310.03684.
>
> [2]: Zhang, Y., Ding, L., Zhang, L., & Tao, D. (2024). Intention analysis makes llms a good jailbreak defender. CoRR abs/2401.06561, 12, 14.
>
> [3]: Sheshadri, A., Ewart, A., Guo, P., Lynch, A., Wu, C., Hebbar, V., ... & Casper, S. (2024). Targeted latent adversarial training improves robustness to persistent harmful behaviors in llms. arXiv e-prints, arXiv-2407.
>
> [4]: The transformer we tested has an input layer, a multi-head attention layer with 4 heads and a key dimension of 64, followed by layer normalization, a dense layer with 128 units, dropout (rate of 0.1), and a final layer normalization. The model then flattens the output, adds another dense layer with 128 units, and concludes with a softmax output layer for classification into 4 classes. The RNN model begins with an input layer, followed by two stacked LSTM layers with 64 units each (the first LSTM layer returns sequences, while the second does not). After the LSTM layers, there is a dense layer with 128 units and a ReLU activation, followed by a softmax output layer for classification into 4 classes. The model is compiled with the Adam optimizer and categorical cross-entropy loss.
>
> [5]: Liu, X., Xu, N., Chen, M., & Xiao, C. (2023). Autodan: Generating stealthy jailbreak prompts on aligned large language models. arXiv preprint arXiv:2310.04451.
>
> [6]: Zeng, Y., Lin, H., Zhang, J., Yang, D., Jia, R., & Shi, W. (2024). How johnny can persuade llms to jailbreak them: Rethinking persuasion to challenge ai safety by humanizing llms. arXiv preprint arXiv:2401.06373.
>
> [7]: Jain, N., Schwarzschild, A., Wen, Y., Somepalli, G., Kirchenbauer, J., Chiang, P. Y., ... & Goldstein, T. (2023). Baseline defenses for adversarial attacks against aligned language models. arXiv preprint arXiv:2309.00614.
>
> [8]: Ji, J., Hou, B., Zhang, Z., Zhang, G., Fan, W., Li, Q., ... & Chang, S. (2024). Advancing the Robustness of Large Language Models through Self-Denoised Smoothing. arXiv preprint arXiv:2404.12274.
>
> [9]: Zhang, Z., Zhang, Q., & Foerster, J. (2024). PARDEN, Can You Repeat That? Defending against Jailbreaks via Repetition. arXiv preprint arXiv:2405.07932.
>
> [10]: Hu, X., Chen, P. Y., & Ho, T. Y. (2024). Gradient cuff: Detecting jailbreak attacks on large language models by exploring refusal loss landscapes. arXiv preprint arXiv:2403.00867.

---

> ### Author Response · Authors · 2024-11-28
> **Kind Reminder: Follow-Up on Author Response**
>
> Thank you once again for taking the time to review our paper and provide such detailed feedback. We deeply value your insights, particularly regarding CurvaLID's architecture, hyperparameter selection, experimental design, and performance on irregular prompts. We kindly ask you to review our Author Response to determine if it sufficiently addresses your concerns. Your thoughtful critique has been invaluable, and we greatly appreciate your efforts.

---

> ### Public Comment · ~Amirali_Abdullah1 · 2026-02-24
> **Somewhat lazy reviewing by u7FH**
>
> If you ask seven questions, and the authors respond to all of them, you should either amend the score or at least acknowledge the effort.

---

### Official Review · Reviewer_YCGd · 2024-11-04

**Soundness:** 2
**Presentation:** 3
**Contribution:** 2
**Rating:** 6
**Confidence:** 4

**Summary:**

The paper proposes an adversarial prompt detection method using the geometric information within the latent representation of an input prompt. Specifically, prompt-level local intrinsic dimensionality using a CNN is proposed as a feature to distinguish the difference between adversarial and benign prompts. TextCurv is also proposed to account for semantic change at the word level, which is another feature to classify adversarial prompts. The proposed method is validated across several adversarial attacks in different LLMs.

**Strengths:**

The paper is clearly written and the proposed method is described clearly.

**Weaknesses:**

1. The size of the data (2500 prompts) is quite limited.

2. Persuasive adversarial prompts [1] will also lead to harmful content but not considered and tested in this work.

3. The proposed detection looks similar to perplexity-based filter. It would be more convincing to include attacks that can bypass perplexity-based filter like AutoDAN [2].

[1] Yi Zeng, Hongpeng Lin, Jingwen Zhang, Diyi Yang, Ruoxi Jia, and Weiyan Shi. 2024. How Johnny Can Persuade LLMs to Jailbreak Them: Rethinking Persuasion to Challenge AI Safety by Humanizing LLMs. In Proceedings of the 62nd Annual Meeting of the Association for Computational Linguistics

[2] Zhu, S., Zhang, R., An, B., Wu, G., Barrow, J., Wang, Z., Huang, F., Nenkova, A. and Sun, T., 2024. AutoDAN: interpretable gradient-based adversarial attacks on large language models. In First Conference on Language Modeling.

**Questions:**

Could the authors address the three weaknesses above?

---

> ### Author Response · Authors · 2024-11-20
> **Author Response (Response to Weakness 1, 2)**
>
> Thanks for your insightful reviews. Please find our response to your questions below:
>
> ---
>
> **Response to Weakness 1**
>
> We used 2,519 prompts for the main results on SOTA benign and adversarial prompts (shown in initial submission Table 1, Page 8). Additionally, we evaluated 1,800 non-English (MultiJail) adversarial prompts (initial submission Table 2, Page 8) and 571 adversarial prompts from PAIR, RandomSearch, and AmpleGCG (initial submission Appendix A.7). In total, we have 3,690 adversarial prompts used in our training and testing.
>
> To the best of our knowledge, this is a substantial set of prompts for testing within this field. We would appreciate it if the reviewer could specifically point out if there are any adversarial prompts datasets we might have missed. We are happy to provide additional experiment results.
>
> The number and diversity of prompts we tested surpass, or at the are comparable to, other SOTA LLM defense studies. For instance:
>
> - **SmoothLLM [1]:** Evaluated its defense with only 4 adversarial prompts (all of which are included in our paper) and no benign prompts. They reported using 100 GCG attacks but did not specify counts for the other 3 adversarial types. In contrast, we tested with 300 GCG adversarial prompts.
>
> - **Gradient Cuff [2]:** Tested 6 adversarial prompt types and 1 benign type, using only 100 prompts per adversarial type, totaling 700 prompts—significantly fewer than the 3,690 adversarial prompts we evaluated. Additionally, they tested with just 100 benign queries, compared to our 1,200 benign prompts.
>
> - **SafeDecoding [3]:** Included 6 adversarial prompt types but specified only 50 prompts for GCG, AutoDAN, and PAIR; 154 prompts for GPTFuzzer-Template; and 330 prompts for HEx-PHI.
>
> - **Latent Adversarial Training[4]:** Evaluated 6 types of adversarial prompts and 3 types of benign prompts, whereas we tested 8 types of adversarial and 4 types of benign prompts.
>
> Our extensive evaluation demonstrates the comprehensiveness of our study compared to existing work.
>
> ---
>
> **Response to Weakness 2**
>
> We are grateful to the reviewer for suggesting an additional persuasive attack [5] for evaluation. We conducted tests on this new attack using 150 adversarial prompts, comprising 50 prompts each targeting GPT-3.5, GPT-4, and Llama2. The experimental setup remains consistent with the details provided in our initial submission (Appendix A.7, Page 18). The experimental results are presented in Table 6 below. CurvaLID successfully detected all persuasive attack prompts. We appreciate the reviewer’s suggestion and have added Table 6 to the revised submission on Page 22.
>
> | Adv. Dataset |  PAIR | RandomSearch | AmpleGCG | Persuasive Attack | AutoDAN | DrAttack |  Avg. |
> |--------------|:-----:|:------------:|:--------:|:-----------------:|:-------:|:--------:|:-----:|
> | Benign Acc.  | 0.973 |       1      |   0.975  |       0.952       |  0.973  |   0.975  | 0.975 |
> | Adv. Acc.    |   1   |       1      |   0.976  |         1         |    1    |     1    | 0.996 |
> | Overall Acc. | 0.983 |       1      |   0.975  |       0.962       |  0.978  |   0.980  | 0.986 |
> | F1 Score     | 0.986 |       1      |   0.987  |       0.974       |  0.986  |   0.987  | 0.985 |
> Table 6: Performance metrics for CurvaLID in PAIR, RandomSearch, AmpleGCG, Persuasive Attack, AutoDAN, and DrAttack.
>
> ---
>
> [1]: Robey, A., Wong, E., Hassani, H., & Pappas, G. J. (2023). Smoothllm: Defending large language models against jailbreaking attacks. arXiv preprint arXiv:2310.03684.
>
> [2]: Hu, X., Chen, P. Y., & Ho, T. Y. (2024). Gradient cuff: Detecting jailbreak attacks on large language models by exploring refusal loss landscapes. arXiv preprint arXiv:2403.00867.
>
> [3]: Xu, Z., Jiang, F., Niu, L., Jia, J., Lin, B. Y., & Poovendran, R. (2024). Safedecoding: Defending against jailbreak attacks via safety-aware decoding. arXiv preprint arXiv:2402.08983.
>
> [4]: Sheshadri, A., Ewart, A., Guo, P., Lynch, A., Wu, C., Hebbar, V., ... & Casper, S. (2024). Targeted latent adversarial training improves robustness to persistent harmful behaviors in llms. arXiv e-prints, arXiv-2407.
>
> [5]: Zeng, Y., Lin, H., Zhang, J., Yang, D., Jia, R., & Shi, W. (2024). How johnny can persuade llms to jailbreak them: Rethinking persuasion to challenge ai safety by humanizing llms. arXiv preprint arXiv:2401.06373.

---

> ### Author Response · Authors · 2024-11-20
> **Author Response (Response to Weakness 3)**
>
> **Response to Weakness 3**
>
> We argue that CurvaLID is fundamentally different from perplexity-based filters. Perplexity-based filters assume that adversarial attacks on LLMs contain gibberish strings that are difficult to interpret, resulting in high perplexity scores [6]. However, this assumption does not hold for SOTA attacks, such as social-engineered attacks, which are highly human-interpretable and often written in grammatically correct and fluent English. Additionally, perplexity-based filters are model-dependent, as their performance varies across different LLMs due to their reliance on the model's measured perplexity.
>
> In contrast, CurvaLID investigates the geometric properties of adversarial prompts without making assumptions about fluency or perplexity. As a result, CurvaLID is capable of detecting both gibberish-based attacks (e.g., GCG, AmpleGCG) and fluently written attacks such as persuasive and social-engineered attacks (e.g., PAIR and DAN). Moreover, CurvaLID is model-agnostic, delivering unified and consistent performance across all LLMs, as it does not depend on the LLM itself for detecting adversarial prompts. We further validated CurvaLID's robustness by experimenting with different word embeddings, showing that it performs effectively across all five tested embeddings (initial submission Table 4, Page 10, Lines 486-497).
>
> We also emphasize that our experiments included attacks that can bypass perplexity-based filters. Specifically, we evaluated social-engineered attacks like PAIR and DAN, which are written in plain, fluent English without gibberish strings. These experiments are detailed in Table 1 (Page 8, Line 378) and Table 6 (Page 19, Line 972) of our initial submission. CurvaLID successfully detected all PAIR and DAN adversarial prompts.
>
> **Regarding comment "It would be more convincing to include attacks that can bypass perplexity-based filter like AutoDAN"**
>
> We thank the reviewer for suggesting an additional adversarial attack, AutoDAN, for experimentation. As shown in Table 6 below, we included 150 AutoDAN prompts, with 50 prompts each targeting Vicuna-7b-v1.5, Llama2-7b, and GPT-4-0613. CurvaLID successfully detected all AutoDAN adversarial prompts.We appreciate the reviewer’s suggestion and have added Table 6 to the revised submission on Page 22.
>
> | Adv. Dataset |  PAIR | RandomSearch | AmpleGCG | Persuasive Attack | AutoDAN | DrAttack |  Avg. |
> |--------------|:-----:|:------------:|:--------:|:-----------------:|:-------:|:--------:|:-----:|
> | Benign Acc.  | 0.973 |       1      |   0.975  |       0.952       |  0.973  |   0.975  | 0.975 |
> | Adv. Acc.    |   1   |       1      |   0.976  |         1         |    1    |     1    | 0.996 |
> | Overall Acc. | 0.983 |       1      |   0.975  |       0.962       |  0.978  |   0.980  | 0.986 |
> | F1 Score     | 0.986 |       1      |   0.987  |       0.974       |  0.986  |   0.987  | 0.985 |
> Table 6: Performance metrics for CurvaLID in PAIR, RandomSearch, AmpleGCG, Persuasive Attack, AutoDAN, and DrAttack.
>
> ---
>
> [6]: Jain, N., Schwarzschild, A., Wen, Y., Somepalli, G., Kirchenbauer, J., Chiang, P. Y., ... & Goldstein, T. (2023). Baseline defenses for adversarial attacks against aligned language models. arXiv preprint arXiv:2309.00614.

---

> > ### Comment · Reviewer_YCGd · 2024-11-25
> >
> > I appreciate the response and would like to increase my score to 6. However, it is still not clear why the content-based detector is helpful for persuasive attacks, as mentioned by other reviewers as well (e.g., lack of thorough analysis), so I won't fight for acceptance.

---

> > > ### Author Response · Authors · 2024-11-28
> > > **Response to the Reviewer's additional feedback**
> > >
> > > We thank the reviewer for their thoughtful feedback and for considering raising their score. We clarified CurvaLID’s use of geometric measures to detect semantic anomalies in persuasive attacks and validated its robustness with tests on reordered prompts, achieving over 96% detection accuracy and 0% ASR.
> > >
> > > ---
> > >
> > > **TextCurv highlights irregular semantic shifts caused by the unnatural connections between words that these attacks exploit, while PromptLID captures the anomalous local intrinsic dimensionality arising from dense clusters of semantic anomalies in persuasive prompts.** Persuasive attacks, such as the PAIR, DAN, and Johnny[1, 2, 3] , rely on subtle manipulations of context and semantics to bypass defenses. These attacks often embed social engineering techniques within the text, making their adversarial nature more covert compared to other attack types. CurvaLID's dual geometric measures—PromptLID and TextCurv—are particularly effective at detecting such prompts. Together, these measures provide a robust framework for identifying the geometric signatures unique to persuasive attacks.
> > >
> > > To validate this, we conducted an experiment to analyze differences in PromptLID and TextCurv between benign and reordered adversarial prompts. Linguistic variations were introduced using GPT-4-o to reorder the words in each sentence of the adversarial prompts while preserving semantic meaning. The experiment involved 100 benign prompts from the Orca, MMLU, AlpacaEval, and TQA datasets, and 100 adversarial prompts from PAIR, DAN, and Persuasive attacks.
> > >
> > > The results, summarized in Table 17 below, demonstrate CurvaLID's effectiveness in distinguishing between benign and adversarial prompts, even after structural reordering. Adversarial prompts exhibited TextCurv values at least 20% higher and PromptLID values over 400% higher than benign prompts. These findings highlight CurvaLID's ability to detect adversarial prompts through their distinctive geometric properties, even under significant structural variations.
> > >
> > > We appreciate the reviewer’s suggestion and have included these results in the revised version of the paper, Appendix A.2.10, Page 25.
> > >
> > > | Geometric measures | TextCurv @ Convolutional Layer 1 |             | TextCurv @ Convolutional Layer 2 |             | PromptLID @ Dense Layer |             |
> > > |--------------------|----------------------------------|-------------|----------------------------------|-------------|-------------------------|-------------|
> > > |                    | Benign                           | Adversarial | Benign                           | Adversarial | Benign                  | Adversarial |
> > > | Avereage Value     | 0.644                            | 0.813(+26.3%)       | 0.341                            | 0.425(+24.6%)       | 3.546                   | 18.223(+413.8%)      |
> > > Table 17: Geometric differences in TextCurv and PromptLID between benign and adversarial prompts. The percentages in parentheses indicate the relative increase in adversarial prompts compared to benign prompts.
> > >
> > > Moreover, we tested CurvaLID on reordered PAIR, DAN, and Johnny prompts (see Table 15 below). Even after applying linguistic variations, the model consistently achieved over 96% detection accuracy across all three attack types. Moreover, it resulted in a 0% attack success rate on Vicuna, meaning that all undetected adversarial prompts failed to bypass the model. These results highlight CurvaLID's robustness against the covert manipulations characteristic of persuasive attacks, establishing it as an effective content-based detector for this challenging category of adversarial prompts. We thank the reviewer for the suggestion and have included Table 15 in our revised version of the paper, A.2.9, Page 25.
> > >
> > > | Adversarial attack                    |  PAIR |  DAN  | Persuasive Attack |
> > > |---------------------------------------|:-----:|:-----:|:-----------------:|
> > > | Benign accuracy                       | 0.951 | 0.971 |       0.966       |
> > > | Adversarial accuracy                  | 0.988 | 1.000 |       0.962       |
> > > | Overall accuracy                      | 0.962 | 0.983 |       0.964       |
> > > | Attack success rate on Vicuna-7B-v1.5 |   0   |   0   |         0         |
> > >
> > > Table 15: Performance of CurvaLID on reordered social-engineered attacks
> > >
> > >
> > > [1]: Chao, P., Robey, A., Dobriban, E., Hassani, H., Pappas, G. J., & Wong, E. (2023). Jailbreaking black box large language models in twenty queries. arXiv preprint arXiv:2310.08419.
> > >
> > > [2]: Shen, X., Chen, Z., Backes, M., Shen, Y., & Zhang, Y. (2023). " do anything now": Characterizing and evaluating in-the-wild jailbreak prompts on large language models. arXiv preprint arXiv:2308.03825.
> > >
> > > [3]:  Yi Zeng, Hongpeng Lin, Jingwen Zhang, Diyi Yang, Ruoxi Jia, and Weiyan Shi. 2024. How Johnny Can Persuade LLMs to Jailbreak Them: Rethinking Persuasion to Challenge AI Safety by Humanizing LLMs. In Proceedings of the 62nd Annual Meeting of the Association for Computational Linguistics

---

### Official Review · Reviewer_Q79u · 2024-11-04

**Soundness:** 1
**Presentation:** 2
**Contribution:** 2
**Rating:** 3
**Confidence:** 5

**Summary:**

Summary:
This paper presents CurvaLID, a framework that claims to detect adversarial prompts in LLMs using geometric measures combining Local Intrinsic Dimensionality (LID) and curvature. While achieving reported empirical results of 99% detection accuracy, the paper suffers from fundamental theoretical flaws, questionable design choices, and significant limitations that are either unexplored or unacknowledged.

Major issues identified:

1. The paper presents three classical curvature definitions (osculating circle, Whewell equation, differential geometric) but implements a formula (dθ/(1/||u|| + 1/||v||)) that has no clear mathematical connection to any of them. The reference to Whewell's equation (dφ/ds) lacks proper theoretical grounding. The authors don't define arc length, don't justify how embedding angles relate to tangential angles, and provide no proof of geometric equivalence to Whewell’s equation. The denominator term using inverse vector norms appears ad hoc and unfortunately comes with no geometric justification.Claims about semantic shifts relating to curvature are made without mathematical rigor or proof. The relationship between their geometric measures and adversarial behavior is assumed (lines 265-266) rather than proven. Therefore, there is a massive disconnect between the theory shown in Section 3 and what is finally implemented.

2. The choice of architecture to use a CNN for text classification is unjustified by the authors. Why choose a CNN when there are more appropriate architectures (like Transformers, RNNs etc) available for sequential data?
The missing aspects are that: (i) there is no justification as to why CNNs would better capture textual properties.
(ii) how are variable length sequences handled by a CNN? Is it via padding?
(iii) How would a CNN capture long-range dependencies in text?
(iv) CNNs typically rely on “spatial order”, what is the equivalent in this setting?

3. In the multi-class classification definition, there is no proper specification given of the label space Q and its relationship to their task. It looks like it’s just a single label per training example.

4. This paper provides no theoretical time and storage complexity analysis of their defense, especially for their key algorithms. While TextCurv computation is O(nd) for sequence length n and embedding dimension d, they don't discuss this in detail. The fact is that if “true differential geometry” metrics were used then the computational complexity is high and mostly polynomial time.

The complexity analysis for critical components should be presented. Such as CNN feature extraction, k-NN computation for PormptLID, Overall pipeline complexity including the MLP classifier. How does complexity scale with longer sequences or larger embedding dimensions or increased batch sizes?

This lack of complexity analysis makes it difficult to assess: i). Practical deployability of their method, ii). scalability to longer prompts, iii). real-time defense capabilities and iv). resource requirements for implementation


5. The method relies critically on exact word order, making it vulnerable to simple paraphrasing. If there are basic linguistic variations like active/passive voice conversion, which retains the semantic meaning, this method would end up with two different representations (geometric measures) completely. This limitation isn’t mentioned or addressed. A simple text restructuring can circumvent this defense.

6. This work presents a limited evaluation in terms of variety of attack types, even when focused on just single-step attacks. There is no analysis of the dataset diversity or potential overlaps between the 8 datasets proposed. There are no evaluations on social engineering attacks (as claimed in introduction) where they stop persuasion attacks for example. There is no testing of non-English attacks despite claims of language-agnostic performance.

7. The CNN feature extraction process is not explained properly and leaves a lot of questions. The hyperparameter choices are arbitrary and do not come with a good justification. It would be nice to see training stability, convergence and sensitivity to initialization studies in the experiments. There must be comparisons made to move naive baseline approaches.

8. There are some reproducibility concerns that must be highlighted too. Critical hyperparameters are left unspecified. Preprocessing details are left incomplete. There is no outline of the training environment and no discussion of potential implementation challenges that are faced.


The paper suffers from multiple fundamental flaws that significantly undermine its contribution. Currently, the paper presents a major disconnect between the presented theoretical framework (which is sophisticated and borrowed from differential geometry) and actual implementation (which is a heuristic). The sequence dependency problem reveals a fundamental limitation that makes the method vulnerable to simple linguistic variations, while the choice of CNN architecture for text processing appears arbitrary and poorly justified.

Additionally, the lack of thorough analysis, missing implementation details, and limited validation make it difficult to assess the method's true effectiveness and reproducibility. The combination of theoretical flaws, practical limitations, and methodological gaps warrants a rejection at this point.

I recommend the following:
1. A complete theoretical overhaul establishing proper connections between differential geometry and their implementation
2. Justification for architectural choices and comparison with alternatives
3. Analysis of and solutions for the sequence dependency problem
4. Comprehensive evaluation across a wider range of linguistic variations and attack types
5. Complete technical details enabling reproducibility

While the empirical results appear promising, they cannot overcome the fundamental theoretical and methodological issues present in the paper.

**Strengths:**

1. The paper introduces an innovative geometric framework (CurvaLID) that combines two complementary measures - Local Intrinsic Dimensionality (LID) and curvature - to detect adversarial prompts. Unlike existing approaches that rely on token-level analysis, the paper presents PromptLID which analyzes geometric differences at the prompt level, and TextCurv which captures semantic shifts at the word level.

2. The paper achieves exceptional performance metrics like (i) Over 0.99 detection accuracy for English prompts,
(ii) 0.994 accuracy for non-English prompts and (iii) Successfully reduces attack success rates to near zero
It also Includes detailed ablation studies showing the importance of both PromptLID and TextCurv components.

3. The solution is highly efficient, requiring only 15 minutes of training on a single Nvidia H100 GPU, compared to competing methods that need up to 16 hours on more extensive hardware setups. The approach is model-agnostic, meaning it doesn't require access to or modification of the underlying LLM. The method also scales well to different languages without requiring retraining.

**Weaknesses:**

1. The paper presents three classical curvature definitions (osculating circle, Whewell equation, differential geometric) but implements a formula (dθ/(1/||u|| + 1/||v||)) that has no clear mathematical connection to any of them. The reference to Whewell's equation (dφ/ds) lacks proper theoretical grounding. The authors don't define arc length, don't justify how embedding angles relate to tangential angles, and provide no proof of geometric equivalence to Whewell’s equation. The denominator term using inverse vector norms appears ad hoc and unfortunately comes with no geometric justification.Claims about semantic shifts relating to curvature are made without mathematical rigor or proof. The relationship between their geometric measures and adversarial behavior is assumed (lines 265-266) rather than proven. Therefore, there is a massive disconnect between the theory shown in Section 3 and what is finally implemented.

2. The choice of architecture to use a CNN for text classification is unjustified by the authors. Why choose a CNN when there are more appropriate architectures (like Transformers, RNNs etc) available for sequential data?
The missing aspects are that: (i) there is no justification as to why CNNs would better capture textual properties.
(ii) how are variable length sequences handled by a CNN? Is it via padding?
(iii) How would a CNN capture long-range dependencies in text?
(iv) CNNs typically rely on “spatial order”, what is the equivalent in this setting?

3. In the multi-class classification definition, there is no proper specification given of the label space Q and its relationship to their task. It looks like it’s just a single label per training example.

4. This paper provides no theoretical time and storage complexity analysis of their defense, especially for their key algorithms. While TextCurv computation is O(nd) for sequence length n and embedding dimension d, they don't discuss this in detail. The fact is that if “true differential geometry” metrics were used then the computational complexity is high and mostly polynomial time.

The complexity analysis for critical components should be presented. Such as CNN feature extraction, k-NN computation for PormptLID, Overall pipeline complexity including the MLP classifier. How does complexity scale with longer sequences or larger embedding dimensions or increased batch sizes?

This lack of complexity analysis makes it difficult to assess: i). Practical deployability of their method, ii). scalability to longer prompts, iii). real-time defense capabilities and iv). resource requirements for implementation


5. The method relies critically on exact word order, making it vulnerable to simple paraphrasing. If there are basic linguistic variations like active/passive voice conversion, which retains the semantic meaning, this method would end up with two different representations (geometric measures) completely. This limitation isn’t mentioned or addressed. A simple text restructuring can circumvent this defense.

6. This work presents a limited evaluation in terms of variety of attack types, even when focused on just single-step attacks. There is no analysis of the dataset diversity or potential overlaps between the 8 datasets proposed. There are no evaluations on social engineering attacks (as claimed in introduction) where they stop persuasion attacks for example. There is no testing of non-English attacks despite claims of language-agnostic performance.

7. The CNN feature extraction process is not explained properly and leaves a lot of questions. The hyperparameter choices are arbitrary and do not come with a good justification. It would be nice to see training stability, convergence and sensitivity to initialization studies in the experiments. There must be comparisons made to move naive baseline approaches.

8. There are some reproducibility concerns that must be highlighted too. Critical hyperparameters are left unspecified. Preprocessing details are left incomplete. There is no outline of the training environment and no discussion of potential implementation challenges that are faced.


The paper suffers from multiple fundamental flaws that significantly undermine its contribution. Currently, the paper presents a major disconnect between the presented theoretical framework (which is sophisticated and borrowed from differential geometry) and actual implementation (which is a heuristic). The sequence dependency problem reveals a fundamental limitation that makes the method vulnerable to simple linguistic variations, while the choice of CNN architecture for text processing appears arbitrary and poorly justified.

Additionally, the lack of thorough analysis, missing implementation details, and limited validation make it difficult to assess the method's true effectiveness and reproducibility. The combination of theoretical flaws, practical limitations, and methodological gaps warrants a rejection at this point.

**Questions:**

I recommend the following:
1. A complete theoretical overhaul establishing proper connections between differential geometry and their implementation
2. Justification for architectural choices and comparison with alternatives
3. Analysis of and solutions for the sequence dependency problem
4. Comprehensive evaluation across a wider range of linguistic variations and attack types
5. Complete technical details enabling reproducibility

While the empirical results appear promising, they cannot overcome the fundamental theoretical and methodological issues present in the paper.

---

> ### Author Response · Authors · 2024-11-20
> **Author Response (Response to Weakness 1 Part 1)**
>
> Thanks for your insightful reviews. Please find our response to your questions below:
>
> ---
> **Response to Weakness 1:**
>
> Thank you for your comment. We will begin by discussing how embedding angles relate to tangential angles to address the reference of TextCurv to Whewell's equation. Our goal is to show that the angle between two word embedding vectors corresponds to the difference in their tangential angles (i.e., the numerator in Whewell's equation). We start by examining the 2D Euclidean plane, where the vector space is two-dimensional. The following proof demonstrates that, in 2D space, the angle between two vectors is equivalent to the difference in their tangential angles.
>
> **Proof: The angle between two vectors is equivalent to the difference in their tangential angles in 2D**
>
> Let $\vec{u}$ and $\vec{v}$ be two vectors in a 2D space, with tangential angles $\phi_u$ and $\phi_v$ relative to the x-axis. Without loss of generality, we consider the positive x-axis represented by the vector $\vec{e_x} = (1, 0)$. We aim to prove that the angle $\theta$ between $\vec{u}$ and $\vec{v}$ is equal to the difference in their tangential angles.
>
> **Step 1: Angle between two vectors $\vec{u}$ and $\vec{v}$**
>
> The angle $\theta$ between two vectors $\vec{u}$ and $\vec{v}$ in Euclidean space is defined by the dot product formula:
>
> $\theta = \arccos \left( \frac{\vec{u} \cdot \vec{v}}{\|\vec{u}\| \|\vec{v}\|} \right).$
>
> **Step 2: Define tangential angles for each vector with respect to the x-axis**
>
> Consider the positive x-axis, represented by the unit vector $\vec{e_x} = (1, 0)$ in 2D space.
>
> - **The tangential angle $\phi_u$ of $\vec{u}$ with respect to $\vec{e_x}$:**
>
> $\phi_u = \arccos \left( \frac{\vec{u} \cdot \vec{e_x}}{\|\vec{u}\|} \right).$
>
> - **The tangential angle $\phi_v$ of $\vec{v}$ with respect to $\vec{e_x}$:**
>
> $\phi_v = \arccos \left( \frac{\vec{v} \cdot \vec{e_x}}{\|\vec{v}\|} \right).$
>
> **Step 3: Express $\vec{u}$ and $\vec{v}$ in terms of their tangential angles**
>
> Using their tangential angles with respect to $\vec{e_x}$, we can write:
>
> $\vec{u} = \|\vec{u}\| \begin{bmatrix} \cos \phi_u\\\sin \phi_u \end{bmatrix}, \quad$
> $\vec{v} = \|\vec{v}\| \begin{bmatrix} \cos \phi_v\\\sin \phi_v \end{bmatrix}.$
>
> **Step 4: Compute the dot product $\vec{u} \cdot \vec{v}$**
>
> Substitute these expressions into the dot product formula:
>
> $\vec{u} \cdot \vec{v} = \|\vec{u}\| \|\vec{v}\| \left( \cos \phi_u \cos \phi_v + \sin \phi_u \sin \phi_v \right).$
>
> Using the trigonometric identity $\cos(\phi_u - \phi_v) = \cos \phi_u \cos \phi_v + \sin \phi_u \sin \phi_v$, we have:
>
> $\vec{u} \cdot \vec{v} = \|\vec{u}\| \|\vec{v}\| \cos(\phi_u - \phi_v).$
>
> **Step 5: Prove the equivalency**
>
> Substitute the result in Step 4 back into the formula for $\theta$:
>
> $\theta = \arccos \left( \frac{\vec{u} \cdot \vec{v}}{\|\vec{u}\| \|\vec{v}\|} \right) = \arccos \left( \cos(\phi_u - \phi_v) \right).$
>
> Since $\arccos(\cos x) = |x|$ for any angle $x$, we have:
>
> $\theta = |\phi_u - \phi_v|.$
>
> We acknowledge that word embedding vectors exist in high-dimensional space, not in 2D. In higher dimensions, defining tangential angles introduces new challenges, as specifying a vector's direction in n-dimensional space requires n−1 angles. However, to explore changes in tangential angles within a text data setting, and given that the angle between two vectors in 2D is equivalent to the difference in their tangential angles, we postulate that for word vectors of consecutive words in text, the angle between their embedding vectors can approximate the change in tangential angle. This approach also allows us to capture semantic shifts effectively even within the limitations of high-dimensional space.

---

> ### Author Response · Authors · 2024-11-20
> **Author Response (Response to Weakness 1 Part 2)**
>
> **Response to Weakness 1 Part 2**
>
> ---
>
> **Relationship between Word Embedding Vector Norms and the Change of Arc Length**
>
> To relate word embedding vector norms and the change of arc length, we rely on an intuitive proportional relationship that aligns with the definitions in Whewell’s equation. We start by considering the relationship between the change in arc length and curvature in the context of Whewell’s equation.
>
> Let $\Delta s$ denote the arc length along a curve and $\kappa$ denote the curvature at a given point on the curve. According to Whewell’s equation, the change in arc length $\Delta s$ is inversely proportional to the curvature $\kappa$ for a fixed change in the tangential angle. Mathematically, this implies:
>
> $\Delta s \propto \frac{1}{\kappa}.$
>
> Intuitively, this means that, for the same change in tangential angle, a smaller arc length $\Delta s$ corresponds to a sharper turn on the curve, which implies a higher curvature $\kappa$.
>
> **Postulating Arc Length in the Context of Semantic Shift**
>
> In the context of semantic shift, we interpret the change in arc length $\Delta s$ as the progression of words within a text. To capture the semantic shift from one word to the next, we approximate $\Delta s$ by measuring the magnitude of the word embedding vectors for consecutive words. Specifically, we denote the magnitudes of two consecutive word embeddings as $\|\vec{u}\|$ and $\|\vec{v}\|$, where $\vec{u}$ and $\vec{v}$ are the embeddings of consecutive words. The use of consecutive words aligns with our approach for evaluating changes in tangential angle, ensuring consistency.
>
> **Relating Arc Length Change to Curvature**
>
> Since the change in arc length is approximated by the magnitude of word embedding vectors, we initially consider $\|\vec{u}\| + \|\vec{v}\|$ as a measure of $\Delta s$. However, directly using $\|\vec{u}\| + \|\vec{v}\|$ in the denominator would contradict the inverse proportionality between $\Delta s$ and curvature $\kappa$. To resolve this, we reinterpret the role of vector magnitudes in the semantic context.
>
> The magnitude of each word embedding vector reflects the semantic "weight" of each word within its context, capturing the tokenizer’s understanding of the word’s significance [1,2]. Consequently, a larger sum $\|\vec{u}\| + \|\vec{v}\|$ suggests a more substantial semantic shift between consecutive words, indicating a sharper change in meaning and thus a higher curvature. Therefore, we assert:
>
> $\kappa \propto \|\vec{u}\| + \|\vec{v}\|.$
>
> To satisfy the inverse relationship between arc length and curvature, we incorporate the inverse of the sum of the word vector magnitudes in our curvature formula. Thus, in our proposed TextCurv metric, we define:
>
> $\text{TextCurv} = \frac{\Delta \theta}{\|\vec{u}\| + \|\vec{v}\|},$
>
> where $\Delta \theta$ represents the change in angle between the consecutive word embeddings $\vec{u}$ and $\vec{v}$, as mentioned above.
>
> ---
>
> Please note that the above explanation was briefly discussed in Section 4.2 in Page 5, Line 238 of our initial submission. We appreciate your suggestion, and we have included the detailed mathematical proof and explanation provided above in the revised version of the paper (Appendix A.1.2, Page 17-19). In future work, we also intend to develop and include more detailed and rigorous proofs on how curvature can be effectively applied in natural language processing.
>
> [1]: Adriaan MJ Schakel and Benjamin J Wilson. Measuring word significance using distributed repre-
> sentations of words. arXiv preprint arXiv:1508.02297, 2015.
>
> [2]: Emily Reif, Ann Yuan, Martin Wattenberg, Fernanda B Viegas, Andy Coenen, Adam Pearce, and
> Been Kim. Visualizing and measuring the geometry of bert. Advances in neural information
> processing systems, 32, 2019.

---

> ### Author Response · Authors · 2024-11-20
> **Author Response to Weakness 2, 3, 4, comment on practical deployability**
>
> **Response to Weakness 2:**
>
> We selected CNN for its computational efficiency and high accuracy within our CurvaLID framework. The variable length sequences handled by a CNN are done by zero-padding. It only requires 15 minutes of training to achieve detection accuracy of 0.992.
>
> We experimented with replacing the CNN architecture with Transformer and RNN models. While both Transformer and RNN achieved comparable detection accuracy (0.98 versus CNN's 0.992), they required almost two to three times longer training times. Details of the Transformer and RNN configurations are provided in Remark [1], with the experimental results presented in Table 13 below:
>
> | Metric                      | CNN (Original Setting) | Transformer |  RNN  |
> |-----------------------------|:----------------------:|:-----------:|:-----:|
> | Detection Accuracy          |          0.992         |    0.984    | 0.989 |
> | Overall Training Time (min) |          14.58         |    39.01    | 25.10 |
>
> Table 13: Comparison of CurvaLID Architectures
>
> In conclusion, the CNN architecture remains the most suitable choice for our CurvaLID model, balancing computational efficiency with exceptional accuracy. We appreciate the reviewer's suggestion to test the Transformer and RNN and have included these experimental results in the revised version of our paper (Table 13 in Appendix A.2.7, Page 24).
>
> ---
>
> **Response to Weakness 3:**
>
> Thank you for your feedback regarding the label space $Q$. In our setup, $Q$ consists of four distinct labels, $\{q_1, q_2, q_3, q_4\}$, each corresponding to one of the four benign datasets we use: Orca, MMLU, AlpacaEval, and TQA (see our paper, Page 7, Line 360–361). Each benign prompt $b$ in the dataset $\mathcal{B}$ is associated with exactly one label $q \in Q$, identifying the specific benign dataset to which the prompt belongs.
>
> This setup enables multi-class classification, where each benign dataset is treated as a separate class. The CNN's objective is to classify each prompt into one of these four classes, optimizing categorical cross-entropy as the loss function. This step is crucial in step 2 of CurvaLID, where we extract the representations of benign and adversarial prompts from this trained CNN (see initial submission, Figure 1, Page 6, Line 270, and Section 4.3, Page 7, Line 325).
>
> ---
>
> **Response to Weakness 4:**
>
> Thank you for the comment. We have included the following time and space complexity analysis in the revised version of our paper, Appendix A.1.3, Page 19.
>
> The time complexity for PromptLID is $\mathcal{O}$$(np)$, where $n$ is the number of prompts and $p$ is the dimensionality of the prompt embeddings. In our implementation, we use the representation layer of the CNN to obtain the embeddings. For TextCurv, the time complexity is $\mathcal{O}$$(nmd)$, where m is the number of words in a prompt and d is the word embedding dimensionality. In our case, we use RoBERTa embeddings for word representations. Therefore, the overall time complexity is $\mathcal{O}$$(n(p+md))$.
>
> The space complexity for PromptLID is $\mathcal{O}$$(np)$, as we store n prompts, each with $p$ dimensions. For TextCurv, the space complexity is $O(nz)$, where $z$ represents the dimensionality of the trained CNN layers used in our computations. Consequently, the space complexity is $\mathcal{O}$$(n(p+z))$.
>
> ---
>
> **Response to "This lack of complexity analysis makes it difficult to assess: i). Practical deployability of their method, ii). scalability to longer prompts, iii). real-time defense capabilities and iv). resource requirements for implementation":**
>
> i) We believe our approach is very capable of being deployed on real-world systems based on the following.
>
> ii) Our default choice, CNN, can handle long-range text sentences. The extracted feature is independent of the text length.
>
> iii) A single forward pass (inference) of our detector takes 6.9 seconds on  a single Nvidia H100 GPU.
>
> iv) We have discussed in the introduction that our detector takes 15 minutes of training on a single Nvidia H100 GPU.
>
> ---
>
> Remark [1]: The transformer we tested has an input layer, a multi-head attention layer with 4 heads and a key dimension of 64, followed by layer normalization, a dense layer with 128 units, dropout (rate of 0.1), and a final layer normalization. The model then flattens the output, adds another dense layer with 128 units, and concludes with a softmax output layer for classification into 4 classes.
> The RNN model begins with an input layer, followed by two stacked LSTM layers with 64 units each (the first LSTM layer returns sequences, while the second does not). After the LSTM layers, there is a dense layer with 128 units and a ReLU activation, followed by a softmax output layer for classification into 4 classes. The model is compiled with the Adam optimizer and categorical cross-entropy loss.

---

> ### Author Response · Authors · 2024-11-20
> **Author Response (Response to Weakness 5)**
>
> **Response to Weakness 5**
>
> We would like to highlight that paraphrasing prompts is, in itself, a form of defense. Defenses based on paraphrasing, such as altering word order or perturbating words and letters, have been explored in numerous studies [2,3,4]. The underlying idea is that adversarial prompts are meticulously designed to exploit vulnerabilities in LLMs, and even minor changes to the prompt can render the attack ineffective.
>
> However, the primary drawback of paraphrasing-based defenses is their model dependency—performance varies significantly across different LLMs—and they inevitably impact the LLM's ability to process benign prompts effectively. In contrast, CurvaLID is model-agnostic and does not alter the content of benign prompts during the defense process.
>
> **Regarding comment "A simple text restructuring can circumvent this defense.":**
>
> We utilized GPT-4-o to reorder the words in every sentence of the adversarial prompts while preserving their semantic meaning. The experimental results, presented in Table 6 below, demonstrate that CurvaLID maintains robust performance, achieving an overall accuracy of 0.984 in detecting these adversarial prompts with altered word order. It is important to note, however, that reordering the words in adversarial prompts may potentially disrupt their effectiveness as attacks, as the content and intent of the original prompts could be compromised. We appreciate the reviewer’s suggestion and have incorporated the corresponding results in Appendix A.2.8 Table 14 (Page 24) of the revised version of our paper.
>
> | **Data class**   | **Accuracy by dataset** |         |             |             | **Accuracy by class** | **Overall accuracy** | **F1 score** |
> |-------------------|-------------------------|---------|-------------|-------------|------------------------|----------------------|--------------|
> |                   | Orca                   | MMLU    | AlpacaEval  | TQA         |                        |                      |              |
> | **Benign**        | 0.993                  | 0.983   | 0.973       | 0.973       | **0.981**              | **0.984**            | **0.984**    |
> |                   | SAP                    | DAN     | MathAttack  | GCG         |                        |                      |              |
> | **Adversarial**   | 0.983                  | 0.963   | 1.000       | 1.000       | **0.987**              |                      |              |
>
> **Table 6:** Performance metrics for CurvaLID on adversarial prompts with reordered word sequences.
>
> ---
>
> [2]: Jain, N., Schwarzschild, A., Wen, Y., Somepalli, G., Kirchenbauer, J., Chiang, P. Y., ... & Goldstein, T. (2023). Baseline defenses for adversarial attacks against aligned language models. arXiv preprint arXiv:2309.00614.
>
> [3]: Robey, A., Wong, E., Hassani, H., & Pappas, G. J. (2023). Smoothllm: Defending large language models against jailbreaking attacks. arXiv preprint arXiv:2310.03684.
>
> [4]: Ji, J., Hou, B., Zhang, Z., Zhang, G., Fan, W., Li, Q., ... & Chang, S. (2024). Advancing the Robustness of Large Language Models through Self-Denoised Smoothing. arXiv preprint arXiv:2404.12274.

---

> ### Author Response · Authors · 2024-11-20
> **Author Response (Response to Weakness 6)**
>
> **Response to Weakness 6**
>
> We would like to point out that we have tested a comparable, if not greater, number and diversity of attacks than other SOTA defenses. Specifically, our evaluation included 8 attacks comprising over 3,000 prompts, whereas SOTA defenses typically evaluate 2–5 attacks with significantly fewer amount of attack prompts [3–7]. The number of attacks and number of prompts tested are mentioned in our initial submission Section 5 (Page 7, Line 337), Section 5.1.1 (Page 7, Line 366), and A.7 (Page 18, Line 958, or revised version A.2.1 on Page 21). For example, SmoothLLM tested four attacks, and PARDEN included only two [3, 5]. Our experiments covered most of the attacks evaluated in existing SOTA defenses.
>
> Regarding the focus on single-step attacks, we note that single-step attacks are the primary scope of research for many SOTA defenses on LLM attacks [2–7]. Other types of attacks often require fundamentally different experimental setups, such as few-shot configurations or additional pre-input instructions, which are beyond the scope of this study. We plan to explore a broader range of attack types in future work.
>
> On the issue of data diversity, we included attacks from a wide range of categories. Specifically:
> - **Gradient-based attacks:** GCG and AmpleGCG, which use adversarial suffixes.
> - **Mathematical attacks:** MathAttack, which targets LLMs' mathematical reasoning abilities.
> - **Adaptive jailbreaking attacks:** RandomSearch, which exploits log probabilities.
> - **Social-engineered attacks:** PAIR and DAN, categorized as persuasion-based attacks. PAIR has been explicitly described and cited as a social engineering attack [9] and is noted as "inspired by social engineering" in its original paper [10]. Similarly, DAN prompts fulfill the criteria for social engineering attacks, employing techniques like manipulation to bypass model restrictions [11–13]. Major strategies, such as prompt injection and privilege escalation, exemplify social engineering.
>
> Given the diverse types of attacks and the distinct algorithms generating them, our dataset exhibits substantial variety with no potential overlap.
>
> Finally, for non-English attacks, we conducted experiments on adversarial prompts in nine different languages, as shown in our initial submission Table 2 (Page 8, Line 396) using the MultiJail dataset. The nine languages are Chinese, Italian, Vietnamese, Arabic, Korean, Thai, Bengali, Swahili, and Javanese. This demonstrates CurvaLID's robustness across multilingual adversarial prompts.
>
> ---
>
> [2]: Jain, N., Schwarzschild, A., Wen, Y., Somepalli, G., Kirchenbauer, J., Chiang, P. Y., ... & Goldstein, T. (2023). Baseline defenses for adversarial attacks against aligned language models. arXiv preprint arXiv:2309.00614.
>
> [3]: Robey, A., Wong, E., Hassani, H., & Pappas, G. J. (2023). Smoothllm: Defending large language models against jailbreaking attacks. arXiv preprint arXiv:2310.03684.
>
> [4]: Ji, J., Hou, B., Zhang, Z., Zhang, G., Fan, W., Li, Q., ... & Chang, S. (2024). Advancing the Robustness of Large Language Models through Self-Denoised Smoothing. arXiv preprint arXiv:2404.12274.
>
> [5]: Zhang, Z., Zhang, Q., & Foerster, J. (2024). PARDEN, Can You Repeat That? Defending against Jailbreaks via Repetition. arXiv preprint arXiv:2405.07932.
>
> [6]: Hu, X., Chen, P. Y., & Ho, T. Y. (2024). Gradient cuff: Detecting jailbreak attacks on large language models by exploring refusal loss landscapes. arXiv preprint arXiv:2403.00867.
>
> [7]: Zhang, Y., Ding, L., Zhang, L., & Tao, D. (2024). Intention analysis prompting makes large language models a good jailbreak defender. arXiv preprint arXiv:2401.06561.
>
> [8]: Turpin, M., Michael, J., Perez, E., & Bowman, S. (2024). Language models don't always say what they think: unfaithful explanations in chain-of-thought prompting. Advances in Neural Information Processing Systems, 36.
>
> [9]: Rababah, B., Kwiatkowski, M., Leung, C., & Akcora, C. G. (2024). SoK: Prompt Hacking of Large Language Models. arXiv preprint arXiv:2410.13901.
>
> [10]: Chao, P., Robey, A., Dobriban, E., Hassani, H., Pappas, G. J., & Wong, E. (2023). Jailbreaking black box large language models in twenty queries. arXiv preprint arXiv:2310.08419.
>
> [11]: Salahdine, F., & Kaabouch, N. (2019). Social engineering attacks: A survey. Future internet, 11(4), 89.
>
> [12]: Salama, R., Al-Turjman, F., Bhatla, S., & Yadav, S. P. (2023, April). Social engineering attack types and prevention techniques-A survey. In 2023 International Conference on Computational Intelligence, Communication Technology and Networking (CICTN) (pp. 817-820). IEEE.
>
> [13]: Ai, L., Kumarage, T., Bhattacharjee, A., Liu, Z., Hui, Z., Davinroy, M., ... & Hirschberg, J. (2024). Defending Against Social Engineering Attacks in the Age of LLMs. arXiv preprint arXiv:2406.12263.

---

> ### Author Response · Authors · 2024-11-20
> **Author Response (Response to Weakness 7, 8)**
>
> **Response to Weakness 7**
>
> To address the reviewer's concerns, we added the following ablation study.
>
> We conducted a study for the CNN hyperparameter selection to determine the optimal architecture based on overall CurvaLID detection accuracy, training time, and stability. Stability was evaluated by measuring the average CurvaLID accuracy across 10 random seeds.  It is important to note that the reported overall accuracy refers specifically to CurvaLID's detection performance, and the running time reflects the total runtime of CurvaLID, not just the CNN component. Furthermore, for each experiment, the tested parameter was the only variable, while all other parameters and settings within CurvaLID remained consistent. The study involved experimenting with the following parameters:
>
> - **Number of convolutional layers:** Tested configurations with 1 to 5 layers.
> - **Activation functions:** Evaluated ReLU, ELU, tanh, sigmoid, and softplus.
> - **Kernel sizes:** Tested kernel sizes ranging from 2 to 5.
> - **Number of parameters:** Adjusted the dense layer sizes to 64, 128, and 256 units.
> - **Epochs:** Tested training with 10, 20, 30, 40,50 epochs.
> - **Batch sizes:** Evaluated sizes of 16, 32, 64, and 128.
> - **Optimizers:** Compared Adam and SGD.
>
> Results can be found in the table below and Table 5 (Appendix A.1.4) in the revised version of our paper. None of these hyperparameters have any significant impact on the detection performance.
>
> We would appreciate it if the reviewer could point out specifically what other hyperparameters should be tested.
>
> |  CNN hyperparameter  | Overall Accuracy | Time (min) |
> |:--------------------:|:----------------:|:----------:|
> |  No. of Conv. Layers |                  |            |
> |           1          |       0.962      |    13.2    |
> |           2          |       0.992      |    14.6    |
> |           3          |       0.992      |    14.6    |
> |           4          |       0.974      |    15.8    |
> |           5          |       0.993      |    15.7    |
> |  Activation function |                  |            |
> |         ReLU         |       0.992      |    14.8    |
> |          ELU         |       0.993      |    15.1    |
> |         tanh         |       0.977      |    15.1    |
> |        Sigmoid       |       0.975      |    14.5    |
> |       Softplus       |       0.991      |    14.8    |
> |      Kernel size     |                  |            |
> |           2          |       0.942      |    14.8    |
> |           3          |       0.992      |    14.9    |
> |           4          |       0.988      |    14.8    |
> |           5          |       0.982      |    15.0    |
> |   Dense layer size   |                  |            |
> |          64          |       0.958      |    14.8    |
> |          128         |       0.990      |    15.2    |
> |          256         |       0.992      |    15.6    |
> |         Epoch        |                  |            |
> |          10          |       0.943      |    14.2    |
> |          20          |       0.991      |    14.9    |
> |          30          |       0.989      |    15.3    |
> |          40          |       0.990      |    15.5    |
> |          50          |       0.988      |    15.3    |
> |      Batch size      |                  |            |
> |          16          |       0.990      |    15.6    |
> |          32          |       0.991      |    14.8    |
> |          64          |       0.988      |    14.7    |
> |          128         |       0.982      |    14.2    |
> |       Optimiser      |                  |            |
> |         Adam         |       0.990      |    15.2    |
> |          SGD         |       0.981      |    15.8    |
> Table 5: Performance metrics for CNN hyperparameter selection.
>
> ---
>
> **Response to Weakness 8**
>
> We would like to point out that we have included the reproducibility statement in Page 11, Lines 540–546 in our initial submission. This statement provides an anonymous GitHub link containing a demo for CurvaLID, along with all critical hyperparameters and pre-processing details.
>
> Furthermore, we have included detailed descriptions of CurvaLID's hyperparameters in the initial version of paper, namely the parameters of the CNN architecture used in CurvaLID Step 1 (Appendix A.2, Page 17) and the MLP architecture used in CurvaLID Step 4 (Appendix A.3, Page 17). Additionally, we have provided the parameters used for generating adversarial prompts (Appendix A.5, Page 17,  and Appendix A.7, Page 18) and the parameters for CurvaLID with One-Class Classification (Appendix A.8, Page 18) in our initial submission.
>
> Regarding the training environment, we have specified in the Introduction (initial submission Page 2, Line 79) that the experiments were conducted using a single Nvidia H100 GPU. Our environment utilized 8 CPU cores and 128 GB RAM. We appreciate the reviewer's suggestion and included additional details in the revised version of the paper (Reproducibility Statement, Page 11).

---

> > ### Comment · Reviewer_Q79u · 2024-11-24
> > **Response to author rebuttal**
> >
> > I appreciate the authors' substantial effort in providing a detailed rebuttal and their willingness to incorporate the suggested modifications into their revised draft. Unfortunately, several fundamental issues persist that warrant maintaining the rejection recommendation:
> >
> > **Theoretical Foundation Gap**:
> > The fundamental theoretical weakness persists. The authors only provide a 2D proof and casually extend it to high-dimensional space without proper justification. Their acknowledgment of challenges in higher dimensions comes without solutions. The inverse vector norm term in their denominator still lacks geometric justification beyond intuitive arguments about semantics.
> >
> > **CNN Architecture Justification**: Their defense of using CNNs focuses solely on speed, ignoring key architectural concerns. They fail to explain how CNNs handle long-range text dependencies, the meaning of spatial order, or why CNNs might work better than purpose-built sequence models. This suggests their architecture choice lacks solid theoretical backing.
> >
> > **Linguistic Variation Problem**: The authors’ response to linguistic variations is problematic on two fronts. While they achieve 0.984 accuracy on GPT-4-reordered prompts, they admit this reordering might disrupt the attacks' effectiveness - suggesting they're detecting broken attacks rather than actual adversarial prompts. Their argument that "paraphrasing is itself a defense" sidesteps the core issue: their method fundamentally depends on word order, making it vulnerable to sophisticated attacks that preserve both semantic meaning and adversarial intent while varying structure. This limitation becomes more severe with increasing prompt length.
> >
> > **Dataset and Evaluation Issues:**
> > While the authors categorize their attacks into different types (gradient-based, mathematical, adaptive jailbreaking, and social-engineered) and test across 9 languages, significant concerns remain. Their claim of "no potential overlap" between datasets lacks empirical validation. They don't justify their sampling strategy or demonstrate how representative their chosen samples are of real-world attacks. Simply testing more attack types than other papers doesn't address the fundamental question of dataset quality and real-world applicability. Their multi-language testing, while broad, needs deeper analysis of how well it captures language-specific attack variations.
> >
> > In short, these unaddressed problems justify rejection. The linguistic variation issue especially shows their method can't handle language's natural complexity. The theoretical gaps and implementation uncertainties further weaken their case.

---

> > > ### Author Response · Authors · 2024-11-28
> > > **Response to the Reviewer's additional feedback (Regarding Foundation Gap, Part 1)**
> > >
> > > We thank the reviewer for their additional feedback and for acknowledging our effort in providing a detailed rebuttal. We appreciate their request for further justification, and the following are our responses to the new comments.
> > >
> > > ---
> > >
> > > **Regarding Foundation Gap:**
> > >
> > > We have included the n-dimensional mathematical proof establishing the relationship between TextCurv and the rate of tangential angular change, along with an extensive geometric justification linking arc length to semantic shifts. We have included these proofs and comprehensive geometric justifications in the revised version of the paper, Appendix A.1.2 (Page 17).
> > >
> > > **Theorem**
> > > For two tangent vectors $\vec{u}$ and $\vec{v}$ in *n*-dimensional Euclidean space, the angle $\theta$ between them is equivalent to the difference in their tangential angles.
> > >
> > >
> > > **Step 1: Angle in *n*-Dimensional Space**
> > >
> > > The angle $\theta$ between two vectors $\vec{u}, \vec{v} \in \mathbb{R}^n$ is defined as:
> > >
> > > $\cos\theta = \frac{\vec{u} \cdot \vec{v}}{\||\vec{u}\|| \||\vec{v}\||},$
> > >
> > > **Step 2: Tangential Angles and Plane Reduction**
> > >
> > > - **Tangential Angles**: Tangential angles describe the orientation of vectors within the specific 2D plane they span. These are defined relative to a chosen reference axis in that plane.
> > >
> > > - **Plane Spanned by** $\vec{u}$ **and** $\vec{v}$: Any two vectors in *n*-dimensional space span a **2D subspace** (a plane). This means the interaction between $\vec{u}$ and $\vec{v}$ (e.g., the angle $\theta$) is fully determined by their projections into this plane.
> > >
> > > - **Orthonormal Basis for the Plane**: Using the Gram-Schmidt process, construct an orthonormal basis {$\{\vec{e}_1, \vec{e}_2\}$}:
> > >
> > >   - Normalize $\vec{u}$ to define $\vec{e}_1$:
> > >     $$
> > >     \vec{e}_1 = \frac{\vec{u}}{\||\vec{u}\||}.
> > >     $$
> > >
> > >   - Define $\vec{e}_2$ as orthogonal to $\vec{e}_1$ and lying in the same plane:
> > >     $$
> > >     \vec{e}_2 = \frac{\vec{v} - (\vec{v} \cdot \vec{e}_1)\vec{e}_1}{\||\vec{v} - (\vec{v} \cdot \vec{e}_1)\vec{e}_1\||}.
> > >     $$
> > >
> > > **Step 3: Expressing $\vec{u}$ and $\vec{v}$ in the Orthonormal Basis**
> > >
> > > In the orthonormal basis $\{\vec{e}_1, \vec{e}_2\}$:
> > > $$
> > > \vec{u} = \||\vec{u}\||\vec{e}_1,
> > > $$
> > > and:
> > > $$
> > > \vec{v} = a\vec{e}_1 + b\vec{e}_2,
> > > $$
> > > where:
> > > $$
> > > a = \vec{v} \cdot \vec{e}_1, \quad b = \vec{v} \cdot \vec{e}_2.
> > > $$
> > >
> > > **Step 4: Computing $\cos\theta$**
> > >
> > > The angle $\theta$ between $\vec{u}$ and $\vec{v}$ is:
> > > $$
> > > \cos\theta = \frac{\vec{u} \cdot \vec{v}}{\||\vec{u}\|| \||\vec{v}\||}.
> > > $$
> > >
> > > Substituting:
> > > $$
> > > \vec{u} \cdot \vec{v} = \||\vec{u}\|| a, \quad \||\vec{v}\|| = \sqrt{a^2 + b^2}.
> > > $$
> > >
> > > Thus:
> > > $$
> > > \cos\theta = \frac{a}{\sqrt{a^2 + b^2}}.
> > > $$
> > >
> > > **Step 5: Computing $\sin\theta$**
> > >
> > > The magnitude of the cross product $\||\vec{u} \times \vec{v}\||$ in the 2D plane is related to $\sin\theta$ by:
> > > $$
> > > \||\vec{u} \times \vec{v}\|| = \||\vec{u}\|| \||\vec{v}\|| \sin\theta.
> > > $$
> > >
> > > Substituting:
> > > $$
> > > \||\vec{u} \times \vec{v}\|| = \||\vec{u}\|| |b|.
> > > $$
> > >
> > > Thus:
> > > $$
> > > \sin\theta = \frac{|b|}{\sqrt{a^2 + b^2}}.
> > > $$
> > >
> > > **Step 6: Computing $\tan\theta$**
> > >
> > > The tangent of $\theta$ is:
> > > $$
> > > \tan\theta = \frac{\sin\theta}{\cos\theta}.
> > > $$
> > >
> > > Substituting:
> > > $$
> > > \tan\theta = \frac{\frac{|b|}{\sqrt{a^2 + b^2}}}{\frac{a}{\sqrt{a^2 + b^2}}}.
> > > $$
> > >
> > > Simplify:
> > > $$
> > > \tan\theta = \frac{|b|}{a}.
> > > $$
> > >
> > > Thus:
> > > $$
> > > \theta = |\arctan(b/a)|.
> > > $$
> > >
> > > **Step 7: Relating $\theta$ to the Tangential Angles**
> > >
> > > In the 2D plane:
> > > - The tangential angle of $\vec{u}$ relative to $\vec{e}_1$ is:
> > >
> > >   $
> > >   \alpha_{\vec{u}} = 0 \quad (\vec{u} \text{ lies entirely along } \vec{e}_1).
> > >   $
> > >
> > > - The tangential angle of $\vec{v}$ is:
> > >   $
> > >   \alpha_{\vec{v}} = \arctan(b/a).
> > >   $
> > >
> > > The difference in tangential angles is:
> > > $
> > > |\alpha_{\vec{u}} - \alpha_{\vec{v}}| = |\arctan(b/a)|.
> > > $
> > >
> > > Thus, the geometric angle $\theta$ between $\vec{u}$ and $\vec{v}$ satisfies:
> > > $
> > > \theta = |\alpha_{\vec{u}} - \alpha_{\vec{v}}|.
> > > $
> > >
> > > This completes the proof. ∎

---

> > > ### Author Response · Authors · 2024-11-28
> > > **Response to the Reviewer's additional feedback (Regarding Architecture Justification)**
> > >
> > > We have included an explanation detailing the suitability of CNNs for capturing local geometric features without necessitating a global sequence modeling approach, while maintaining simplicity and high accuracy. We want to emphasize that our experimental results do not preclude the use of RNNs or transformers in CurvaLID; however, using CNNs provides better computational efficiency.
> > >
> > > Our selection of CNNs was guided by their capability to efficiently extract local features, which aligns with our focus on capturing geometric properties such as PromptLID and TextCurv. These measures emphasize local intrinsic dimensions and curvature at prompt-level and word-level representations respectively.
> > >
> > > While it is true that CNNs are not explicitly designed for modeling long-range dependencies, our problem does not necessitate a global sequence modeling approach. Instead, adversarial prompts exhibit local geometric anomalies that CNNs are well-suited to detect. Spatial order in text is represented as sequential relationships in word embeddings, and CNNs effectively encode these relationships.
> > >
> > > Compared to purpose-built sequence models like transformer and RNN, CNNs offer computational simplicity without compromising accuracy for our task. Our experimental results (revised version of paper, A.2.7, Page 24, also shown below in Table 13) show that CNN maintained a high detection accuracy but also reducing the overall computation time for CurvaLID when compared with using RNN and transformer. Furthermore, the complementary design of PromptLID and TextCurv mitigates any limitations in modeling long-range dependencies by focusing on localized features.
> > >
> > > | Metric                      | CNN (Original Setting) | Transformer |  RNN  |
> > > |-----------------------------|:----------------------:|:-----------:|:-----:|
> > > | Detection Accuracy          |          0.992         |    0.984    | 0.989 |
> > > | Overall Training Time (min) |          14.58         |    39.01    | 25.10 |
> > >
> > > Table 13: Comparison of CurvaLID Architectures

---

> ### Author Response · Authors · 2024-11-28
> **Response to the Reviewer's additional feedback (Regarding Foundation Gap, Part 2)**
>
> Now we investigate the relationship between word embedding vector norms and the change of arc length. We start with the goal of approximating the arc length $ \Delta s $ between two consecutive word embeddings, $ \vec{u} $ and $ \vec{v} $, in a high-dimensional space. Arc length is classically defined as the integral of the norm of the tangent vector along the curve. For discrete data points, this is approximated as a sum of the Euclidean distances between points.
>
> **1. Discrete Approximation of Arc Length**
>
> Given consecutive embeddings $ \vec{u} $ and $ \vec{v} $, the arc length between these points can be approximated as:
>
> $ \Delta s = \||\vec{u} - \vec{v}\||. $
>
> However, directly using $ \||\vec{u} - \vec{v}\|| $ would treat the embeddings purely as geometric points and ignore their semantic significance as encoded by the vector magnitudes.
>
> **2. Semantic Weight and Embedding Norms**
>
> In NLP, the norm of a word embedding, $ \||\vec{u}\|| $, encodes the semantic "weight" or importance of a word within its context [1,2]. Larger norms indicate that the embedding carries more semantic information, while smaller norms suggest less significance.
>
> For two consecutive embeddings, $ \vec{u} $ and $ \vec{v} $, their combined semantic importance is proportional to their norms:
>
> $ \text{Semantic Importance} \propto \||\vec{u}\|| + \||\vec{v}\||. $
>
> **3. Inverse Proportionality and Arc Length**
>
> To align with the geometric principle in Whewell’s equation that relates curvature ($ \kappa $) and arc length ($ \Delta s $) as:
>
> $ \kappa \propto \frac{1}{\Delta s}, $
>
> we posit that arc length ($ \Delta s $) should *decrease* when the semantic importance ($ \||\vec{u}\|| + \||\vec{v}\|| $) increases.
>
> This motivates the choice of the *inverse relationship*:
>
> $ \Delta s \propto \frac{1}{\||\vec{u}\|| + \||\vec{v}\||}. $
>
> **4. Sum of Inverse Norms as Arc Length**
>
> While $ \|\vec{u}\| + \|\vec{v}\| $ represents the combined semantic importance of two embeddings, directly using it in the denominator would contradict the inverse proportionality between $ \Delta s $ and $ \kappa $. Instead, we take the *inverse of the norms individually*, which ensures the arc length is smaller for larger semantic weights.
>
> Thus, the arc length approximation becomes:
>
> $ \Delta s \propto \frac{1}{\||\vec{u}\||} + \frac{1}{\||\vec{v}\||}. $
>
> The reasoning is that embeddings with larger norms (higher semantic significance) should have smaller contributions to the overall arc length, reflecting the sharper semantic transitions between significant words.
>
>
> [1]: Adriaan MJ Schakel and Benjamin J Wilson. Measuring word significance using distributed representations of words. arXiv preprint arXiv:1508.02297, 2015.
>
> [2]: Emily Reif, Ann Yuan, Martin Wattenberg, Fernanda B Viegas, Andy Coenen, Adam Pearce, and Been Kim. Visualizing and measuring the geometry of bert. Advances in neural information
> processing systems, 32, 2019.

---

> ### Author Response · Authors · 2024-11-28
> **Response to the Reviewer's additional feedback (Regarding Linguistic Variation problem, Part 1)**
>
> We have included evidence of CurvaLID's robustness against reordered and paraphrased adversarial prompts, and an explanation of its resilience to structural variations and prompt length.
>
> Below, we address the specific concerns:
>
> **1. Effectiveness Against Reordered Prompts:**
>
> * As noted in defenses like SmoothLLM, not all attacks are rendered ineffective after reordering word order [1]. For instance, the PAIR attack, a social-engineered persuasive attack, retained 46% and 24% attack success rates on Vicuna and GPT-4, respectively, after undergoing SmoothLLM defenses.
> * In contrast, we tested CurvaLID on reordered PAIR attack prompts and achieved over 96% detection accuracy, resulting in a 0% attack success rate on Vicuna (i.e., all undetected adversarial prompts failed to jailbreak Vicuna). To further validate this, we evaluated CurvaLID on reordered prompts from DAN and Persuasive attacks [3,4], both of which, like PAIR, are social-engineered persuasive attacks that can preserve both semantic meaning and adversarial intent while varying structure. CurvaLID consistently achieved over 96% detection accuracy across all three attack types and maintained a 0% attack success rate on Vicuna. These results demonstrate that our method is robust even against sophisticated, linguistically varied prompts specifically crafted to bypass defenses. All of these results are presented in Table 15 below and included in our revised version of the paper, A.2.9, Page 25.
>
>
>
> | Adversarial attack                    |  PAIR |  DAN  | Persuasive Attack |
> |---------------------------------------|:-----:|:-----:|:-----------------:|
> | Benign accuracy                       | 0.951 | 0.971 |       0.966       |
> | Adversarial accuracy                  | 0.988 | 1.000 |       0.962       |
> | Overall accuracy                      | 0.962 | 0.983 |       0.964       |
> | Attack success rate on Vicuna-7B-v1.5 |   0   |   0   |         0         |
>
> Table 15: Performance of CurvaLID on reordered social-engineered attacks
>
> **2. Relevance of Paraphrasing as a Defense:**
> * Our statement that "paraphrasing is itself a defense" is a factual observation supported by prior works, as many SOTA adversarial prompts fail to maintain their effectiveness under paraphrasing defenses [1,2].
> * To ensure completeness, we invite the reviewer to suggest adversarial prompts that preserve both semantic meaning and adversarial intent while varying structure. We are eager to test our method against such cases to further validate its robustness.

---

> ### Author Response · Authors · 2024-11-28
> **Response to the Reviewer's additional feedback (Regarding Linguistic Variation problem, Part 2)**
>
> **3. Structural Variations and Word Order:**
>
> * CurvaLID effectively captures the geometric properties of adversarial prompts, even under structural variations. While it leverages word order to analyze local geometric patterns, this does not imply a fundamental vulnerability to structural changes. Adversarial prompts inherently exhibit irregular geometric characteristics, including higher curvature and Local Intrinsic Dimensionality (LID) at the prompt level, irrespective of word order. The complementary geometric measures employed by CurvaLID—PromptLID and TextCurv—are designed to detect these anomalies, enabling reliable identification across a variety of structural transformations. Sophisticated attacks that preserve adversarial intent and meaning while varying structure still exhibit detectable geometric anomalies, making them identifiable by our method.
> * To further validate the robustness of CurvaLID, we conducted an additional experiment to analyze PromptLID and TextCurv differences between benign and adversarial prompts after reordering the adversarial prompts. Linguistic variations were introduced by using GPT-4-o to reorder the words in each sentence of the adversarial prompts while preserving semantic meaning. The experiment included 100 benign prompts from the Orca, MMLU, AlpacaEval, and TQA datasets, and 100 adversarial prompts from PAIR, DAN, and Persuasive attacks.
> * The results, summarized in Table 17 below, demonstrate the robustness of CurvaLID in distinguishing between benign and adversarial prompts even after reordering the adversarial prompts. Adversarial prompts exhibited TextCurv values at least 20% higher than benign prompts and PromptLID values over 400% higher than their benign counterparts. These findings highlight the ability of CurvaLID to detect adversarial prompts through their distinctive geometric properties, even under significant structural variations.
> * We thank the reviewer for the advice given and we have included the above result in our revised version of the paper, A.2.10, Page 25.
>
> | Geometric measures | TextCurv @ Convolutional Layer 1 |             | TextCurv @ Convolutional Layer 2 |             | PromptLID @ Dense Layer |             |
> |--------------------|----------------------------------|-------------|----------------------------------|-------------|-------------------------|-------------|
> |                    | Benign                           | Adversarial | Benign                           | Adversarial | Benign                  | Adversarial |
> | Average Value     | 0.644                            | 0.813(+26.3%)       | 0.341                            | 0.425(+24.6%)       | 3.546                   | 18.223(+413.8%)      |
> Table 17: Geometric differences in TextCurv and PromptLID between benign and adversarial prompts. The percentages in parentheses indicate the relative increase in adversarial prompts compared to benign prompts.
>
> **4. Prompt Length:**
>
> * While longer prompts might theoretically exacerbate certain limitations, our empirical results on datasets with varying prompt lengths show no significant degradation in detection accuracy. In our experiments, prompt lengths ranged from as few as 5 words to as many as 1973 words, with an average length of 98.26 words and a median of 35 words. Despite this significant variability (standard deviation of 156.26 words), CurvaLID consistently demonstrated robust performance. These findings suggest that the geometric properties analyzed by CurvaLID scale effectively with prompt length, even under extreme variations.
>
> [1]: Robey, A., Wong, E., Hassani, H., & Pappas, G. J. (2023). Smoothllm: Defending large language models against jailbreaking attacks. arXiv preprint arXiv:2310.03684.
>
> [2]: Jain, N., Schwarzschild, A., Wen, Y., Somepalli, G., Kirchenbauer, J., Chiang, P. Y., ... & Goldstein, T. (2023). Baseline defenses for adversarial attacks against aligned language models. arXiv preprint arXiv:2309.00614.
>
> [3]: Shen, X., Chen, Z., Backes, M., Shen, Y., & Zhang, Y. (2023). " do anything now": Characterizing and evaluating in-the-wild jailbreak prompts on large language models. arXiv preprint arXiv:2308.03825.
>
> [4]:  Yi Zeng, Hongpeng Lin, Jingwen Zhang, Diyi Yang, Ruoxi Jia, and Weiyan Shi. 2024. How Johnny Can Persuade LLMs to Jailbreak Them: Rethinking Persuasion to Challenge AI Safety by Humanizing LLMs. In Proceedings of the 62nd Annual Meeting of the Association for Computational Linguistics

---

> ### Author Response · Authors · 2024-11-28
> **Response to the Reviewer's additional feedback (Regarding Dataset and Evaluation Issues, Part 1)**
>
> We validated no dataset overlap, demonstrated CurvaLID’s 0.982 accuracy on unseen datasets, reduced adversarial prompts' ASR to zero in real-world tests, and showed its robustness across nine languages with consistent geometric signatures.
>
> Below, we address the specific concerns:
>
> **1. Dataset Overlap:**
>
> There is no dataset overlap, as we have done experiment checking if there are duplicate prompts along all tested prompts.
>
> The datasets used in our experiments are generated by different algorithms from distinct sources in the literature, and the jailbreaking methods are fundamentally different across these datasets.
>
> To ensure there is no overlap, we conducted an empirical test to identify any identical prompts across datasets, and we confirm that there are no duplicate prompts. This validation will be explicitly included in the revised manuscript.
>
> **2.  Sampling strategy:**
>
> We would like to clarify that CurvaLID employs a train-test split strategy, using 80% of the data for training and 20% for testing. Importantly, there is no overlap among different benign datasets, and there is no overlap within individual datasets.
>
> To further address the concern, we conducted an ablation study by training CurvaLID on two benign datasets and testing it on the remaining two. Specifically, we trained CurvaLID using only Orca and MMLU as benign data and evaluated it on AlpacaEval and TQA. The results, shown in Table 17, demonstrate an overall detection accuracy of 0.982, just one percentage point lower than when trained on all four benign datasets. These findings indicate that CurvaLID's performance remains robust and is not overly optimistic, even when tested on unseen benign datasets. We thank the reviewer for their insightful comment and have incorporate the above discussion into the revised version of the paper, Table 17 in Appendix A.2.11, Page 25.
>
> | **Data class**    | **AlpacaEval** | **TQA**   |  |    |  | **Accuracy by class** | **Overall accuracy** | **F1 score** |
> |--------------------|----------------|-----------|---------|-------------------------|---------|------------------------|-----------------------|--------------|
> | **Benign**         | 0.963         | 0.9194    |         |         |                | **0.9412**       | **0.982**            |       **0.98**     |    |
> |                    |  **SAP**  |  **DAN** | **MathAttack** | **GCG**  |    |                        |                       |              |
> | **Adversarial**    |      1          |    1       | 1       | 1       |                    |    **1**              |                       |              |
>
> **Table 17:** CurvaLID with different training and testing benign datasets.
>
> **3. Applicability to Real-World Scenarios:**
>
> CurvaLID effectively blocks nearly all attacks on Vicuna in real-world scenarios. In our initial submission, we conducted experiments, which are discussed in Section 5 (Lines 343-344), and reported the performance of CurvaLID in a real-world application on Vicuna (A.12, Page 21). As indicated in Table 19 (Page 24 of the initial submission), CurvaLID successfully reduced the ASR of most attacks to zero, outperforming the SOTA defenses that were studied. In the revised version of the paper, we further included CurvaLID’s performance on three persuasive social-engineered attacks, namely PAIR, DAN, and Johnny [1,2,3]. The results, summarized in Table 17 above, demonstrate the robustness of CurvaLID in distinguishing between benign and adversarial prompts. We have included the experimental result in our revised version of the paper, A.2.10, Page 25.
>
>
> [1]: Chao, P., Robey, A., Dobriban, E., Hassani, H., Pappas, G. J., & Wong, E. (2023). Jailbreaking black box large language models in twenty queries. arXiv preprint arXiv:2310.08419.
>
> [2]: Shen, X., Chen, Z., Backes, M., Shen, Y., & Zhang, Y. (2023). " do anything now": Characterizing and evaluating in-the-wild jailbreak prompts on large language models. arXiv preprint arXiv:2308.03825.
>
> [3]:  Yi Zeng, Hongpeng Lin, Jingwen Zhang, Diyi Yang, Ruoxi Jia, and Weiyan Shi. 2024. How Johnny Can Persuade LLMs to Jailbreak Them: Rethinking Persuasion to Challenge AI Safety by Humanizing LLMs. In Proceedings of the 62nd Annual Meeting of the Association for Computational Linguistics

---

> > ### Author Response · Authors · 2024-11-28
> > **Response to the Reviewer's additional feedback (Regarding Dataset and Evaluation Issues, Part 2)**
> >
> > **4. Dataset Quality:**
> >
> > All tested adversarial prompts are effective in real-world scenarios. As discussed in their respective papers, all the attacks we researched are applicable to real-world scenarios. Additionally, some of the attacks were generated by us through their Github repositories, such as GCG, ampleGCG, and PAIR; these are successful attacks on LLMs that can jailbreak LLMs in the current market. Furthermore, DAN attacks are directly sourced from online platforms, as they are considered "wild attacks." These attacks have been successfully generated by users and shared on forums like Reddit, Discord, and Twitter. Therefore, we can assure their real-world applicability. We are eager to conduct additional tests on adversarial attacks if the reviewer has specific suggestions to offer.
> >
> > **5. Multi-language testing:**
> >
> > The nine languages tested in our experiments represent a diverse set of linguistic families and structures. It is important to note that adversarial prompts across languages still exhibit consistent geometric signatures, such as higher curvature, which CurvaLID is designed to detect. This consistency indicates that our method captures fundamental properties of adversarial prompts, regardless of language.
> >
> > We emphasize that the number of multi-language adversarial prompts tested, both in terms of languages and the number of prompts, is consistent with other SOTA defenses [1,2]. Our experimental setup and dataset coverage align with established standards. We are also open to testing CurvaLID on additional datasets, should the reviewer have specific suggestions.
> >
> > [1]: Zhang, Y., Ding, L., Zhang, L., & Tao, D. (2024). Intention analysis makes llms a good jailbreak defender. CoRR abs/2401.06561, 12, 14.
> >
> > [2]: Wang, X., Wu, D., Ji, Z., Li, Z., Ma, P., Wang, S., ... & Rahmel, J. (2024). SelfDefend: LLMs Can Defend Themselves against Jailbreaking in a Practical Manner. arXiv preprint arXiv:2406.05498.

---

### Official Review · Reviewer_AkuS · 2024-11-11

**Soundness:** 3
**Presentation:** 2
**Contribution:** 3
**Rating:** 6
**Confidence:** 2

**Summary:**

This paper studies adversarial prompt detection. It revitalizes metrics of LID from anomaly detection to this task, by estimating the score on a learned prompt-level representation instead of a suboptimal word-level one. The paper further proposes TextCurve score, and merges these two metrics into a single score for the final decision. The final binary classifier needs to be trained on a collect. Extensive empirical results and ablation study demonstrate the generation and robustness of the proposed metrics despite the need for training.

**Strengths:**

1. Novelty: The author revitalized LID score in detecting adversarial prompts by estimating it on the learned aggregated output of word level embeddings. It further proposes TextCurve to capture the extra curvature information around the prompt. The final decision takes account for both metrics via a learned linear combination.
2. Extensive experimental evaluations. The paper includes a detailed amount of empirical evaluations and analysis to support the effectiveness of the proposed method, as well as providing insights into how different components affect its performance.

**Weaknesses:**

1. How is curveLID’s defenses against jailbreaking algorithms rather than static harmful prompt? It Whitebox methods are not applicable in this setting, but the author can potentially try some blackbox methods, such as DrAttack or Multijail? It seems that the authors include several defense methods in the Appendix, but the experiment seems not directly related to the proposed metrics (I could be wrong).
2. Minor: the appendix is very rich in useful information so it cannot be ignored. Tut currently the organization is quite flattened so a bit difficult to track everything. Organizing it into sections/subsections (e.g. all ablations are grouped together, all analysis are grouped together) might improve its readability.

**Questions:**

1. Several parts of the CurveLID metric requires fitting on training data of adversarial v.s. benign prompts. I’d expect that the performance will depend heavily on the coverage of training data distribution, but judging from the experimental results it seems the generalization is pretty descent, have the authors had any insights on why this is the case?

---

> ### Author Response · Authors · 2024-11-20
> **Author Response**
>
> Thanks for your insightful reviews. Please find our response to your questions below:
>
> ---
> **Response to Weakness 1:**
>
> CurvaLID is designed to identify harmful prompts generated by jailbreaking algorithms, enabling it to reject these prompts before they can interact with or exploit the model.
>
> We would like to clarify that our paper encompasses both white-box and black-box attacks. MultiJail, mentioned by the reviewer, has already been examined in our initial submission. The results are in Table 2 on Page 8 in our initial submission. We have also included black-box attacks in SAP, DAN, MathAttack, MultiJail, PAIR, and RandomSearch, and white-box attacks in GCG and AmpleGCG in our initial submission. Their results can be found in the following tables: Table 1 on Page 8 (for SAP, DAN, MathAttack, GCG); Table 2 on Page 8(for MultiJail); and Table 6 on Page 19 (for PAIR, RandomSearch, and AmpleGCG).
>
> We have conducted extensive experiments incorporating DrAttack, as detailed in Table 6 below. This evaluation included 150 DrAttack prompts, with 50 prompts targeting each of Vicuna-7B, LLaMA-2-7B-Chat, and GPT-3.5-turbo. The experimental setup remains consistent with the details provided in our initial submission (Appendix A.2.1, Page 21). From the results, CurvaLID successfully detected all DrAttack prompts with high accuracy. We appreciate the reviewer’s suggestion and have added Table 6 to the revised submission in Appendix A.2.1 Page 22.
>
> | Adv. Dataset |  PAIR | RandomSearch | AmpleGCG | Persuasive Attack | AutoDAN | DrAttack |  Avg. |
> |--------------|:-----:|:------------:|:--------:|:-----------------:|:-------:|:--------:|:-----:|
> | Benign Acc.  | 0.973 |       1      |   0.975  |       0.952       |  0.973  |   0.975  | 0.975 |
> | Adv. Acc.    |   1   |       1      |   0.976  |         1         |    1    |     1    | 0.996 |
> | Overall Acc. | 0.983 |       1      |   0.975  |       0.962       |  0.978  |   0.980  | 0.986 |
> | F1 Score     | 0.986 |       1      |   0.987  |       0.974       |  0.986  |   0.987  | 0.985 |
> Table 6: Performance metrics for CurvaLID in PAIR, RandomSearch, AmpleGCG, Persuasive Attack, AutoDAN, and DrAttack.
> ---
> **Response to Weakness 2:**
>
> Thank you for your valuable feedback. In the revised version of the paper, we have reorganized the appendix into 4 sections, namely supplementary information about CurvaLID, ablation studies, supplementary information on LID analysis, and performance of other SOTA defenses.
>
> ---
> **Response to Question 1:**
>
> We believe the generalization results are due to the PromptLID and TextCurv features exhibiting a clear and substantial distinction between benign and adversarial prompts, as shown in Figure 2(c) and Table 4 in our initial submission.
>
> We have also included the experiment with a one-class classification setup, where defenders only have access to benign prompts for training. We used outlier detection methods such as Local Outlier Factor (LOF) and isolation forest. Results are in Tables 7 and 8 in our initial submission, which show that these methods achieved comparable detection accuracy of around 0.9. This further demonstrates that the PromptLID and TextCurv features are very distinctive.
>
> ---

---

> ### Author Response · Authors · 2024-11-28
> **Kind Reminder: Follow-Up on Author Response**
>
> Thank you once again for taking the time to review our paper and for providing such detailed and thoughtful feedback. We greatly value your insights, particularly regarding testing on new adversarial attacks and reformatting the appendix. We kindly invite you to review our Author Response to see if it sufficiently addresses your concerns. Your thoughtful critique has been immensely helpful, and we sincerely appreciate your time and effort.

---

### Meta-Review · Area_Chair_Zf1S · 2024-12-18

**Metareview:**

Adversarial prompts that LLM pose significant challenges to their safe deployments. The authors propose CurvaLID, a unified adversarial prompt detection algorithm that combines two geometric measures: Local Intrinsic Dimensionality for prompt-level analysis and TextCurv for word-level semantic shifts via curvature. CurvaLID achieves over 0.99 detection accuracy, reducing adversarial attack success rates and providing consistent performance across diverse prompts and LLMs.

Most reviewers acknowledge the paper's clarity and the efficiency & efficacy of the proposed methods. However, most concerns lie in limited innovation, theoretical analysis, and more diverse experimental validation on different testing scenarios. We recommend a reject.

**Additional Comments On Reviewer Discussion:**

During the rebuttal period, the authors did a good job and conducted great efforts in the discussion. We believe that incorporating the discussions accordingly will further strength future revisions.

---

### Decision · Program_Chairs · 2025-01-22

Reject